# Age and gravitational separation of the stratospheric air over Indonesia

Satoshi Sugawara[1], Shigeyuki Ishidoya[2], Shuji Aoki[3], Shinji Morimoto[3], Takakiyo Nakazawa[3], Sakae Toyoda[4], Yoichi Inai[5, 3], Fumio Hasebe[5], Chusaku Ikeda[6], Hideyuki Honda[6], Daisuke Goto[7], and Fanny A. Putri[8]

[1]Miyagi University of Education, Sendai 980-0845, Japan
[2]National Institute of Advanced Industrial Science and Technology (AIST), Tsukuba 305-8569, Japan
[3]Center for Atmospheric and Oceanic Studies, Tohoku University, Sendai 980-8578, Japan
[4]Tokyo Institute of Technology, Yokohama 226-8502, Japan
[5]Faculty of Environmental Earth Science, Hokkaido University, Sapporo, 060-0810, Japan
[6]Institute of Space and Astronautical Science, Japan Aerospace Exploration Agency, Sagamihara 252-5210, Japan.
[7]National Institute of Polar Research, Tokyo 190-8518, Japan
[8]National Institute of Aeronautics and Space, Bandung 40173, Indonesia

*Correspondence to*: Satoshi Sugawara (sugawara@staff.miyakyo-u.ac.jp)

**Abstract.** The gravitational separation of major atmospheric components, in addition to the age of air, would provide additional useful information about the stratospheric circulation. However, observations of the age of air and gravitational separation are still geographically sparse, especially in the tropics. In order to address this issue, air samples were collected over Biak, Indonesia in February 2015 using four large plastic balloons, each loaded with two compact cryogenic samplers. With a vertical resolution of better than 2 km, air samples from seven different altitudes were analyzed for $CO_2$ and $SF_6$ mole fractions, $\delta^{15}N$ of $N_2$, $\delta^{18}O$ of $O_2$, and $\delta(Ar/N_2)$, to examine vertically dependent age and gravitational separation of air in the Tropical Tropopause Layer (TTL) and the equatorial stratosphere. By comparing their measured mole fractions with aircraft observation in the upper tropical troposphere, we have found that $CO_2$ and $SF_6$ ages increase gradually with increasing altitude from the TTL to 22 km, and then rapidly from there up to 29 km. The $CO_2$ and $SF_6$ ages agree well with each other in the TTL and in the lower stratosphere, but show a significant difference above 24 km. The average values of $\delta^{15}N$ of $N_2$, $\delta^{18}O$ of $O_2$, and $\delta(Ar/N_2)$ all show small but distinct upward decrease due to the gravitational separation effect. Simulations with a two-dimensional atmospheric transport model indicate that the gravitational separation effect decreases as tropical upwelling is enhanced. From the model calculations with enhanced eddy mixing, it is also found that the upward increase in air age is magnified by horizontal mixing. These model simulations also show that the gravitational separation effect remains relatively constant in the lower stratosphere. The results of this study strongly suggest that the gravitational separation, combined with the age of air, can be used to diagnose air transport processes in the stratosphere.

## 1 Introduction

The mean age of air, defined as the mean transit time of air parcels in the stratosphere, provides important information

about various stratospheric transport processes. It is expected that possible long-term changes in the Brewer–Dobson (BD) circulation are detectable from changes in the mean age of air evaluated from measurements of the stratospheric $CO_2$ and $SF_6$ mole fractions. Balloon and satellite observations (Engel et al., 2009, 2017; Stiller et al., 2012) found that the age of air derived from $CO_2$ and $SF_6$ in the stratosphere over the northern mid-latitudes did not show any significant trend above 25 km

over the last 30 years, although satellite observations have indicated that the $SF_6$ age might have increased for the period 2002–2010. To further add fire to the debate, results obtained from balloon and satellite observations do not agree with the recent model predictions that the BD circulation is accelerated due to an enhancement of mass flux from the tropical troposphere into the stratosphere (Austin and Li, 2006; Li et al., 2008). More recently, Haenel et al. (2015) reported a negative trend in the age of air in the lowermost tropical stratosphere, by revising the MIPAS/$SF_6$ age data. Bönisch et al.

(2011) suggested that the increased upwelling in the tropics after 2000 enhanced the lower stratospheric transport from the tropics into the extra-tropics. From an analysis of the ERA-Interim dataset, Diallo et al. (2012) also showed a negative trend over the 1989–2010 period in the lower stratosphere below 25 km. Linz et al. (2017) discussed the strength of the meridional overturning circulation of the stratosphere by using satellite observation data of $SF_6$ and $N_2O$, and suggested that a mesospheric $SF_6$ loss is important for age estimation using $SF_6$ mole fraction in the upper layer.

A clock tracer, such as $CO_2$ or $SF_6$, is lifted up from the tropical upper troposphere into the stratosphere and is transported to mid-latitudes; part of this then returns to the lower latitudes via mixing process in the stratosphere, which results in increasing the clock tracer age in the lower latitudes. Therefore, to discuss a change in the mean age estimated using the clock tracer, it is important to separately evaluate the respective effects of mean circulation and mixing processes on the air age (Garny et al., 2014; Ploeger et al., 2015; Linz et al., 2016). Recently, it was shown by high-precision measurements of

the isotopic ratios of $N_2$, $O_2$, and Ar and the Ar/$N_2$ ratio that the gravitational separation of the air is clearly observable in the northern mid-latitude stratosphere (Ishidoya et al., 2006, 2008a, 2008b, 2013). Ishidoya et al. (2013) also reported that the gravitational separation could be used as a new indicator to detect long-term behavior of the BD circulation.

Most stratospheric air originates in the Tropical Tropopause Layer (TTL) and is then slowly lifted up from lower to upper levels. Therefore, observations of stratospheric air compositions, from which the age and gravitational separation of air are

deduced, are necessary in the equatorial region to understand the various transport processes in the stratosphere. Collecting whole air with balloon-borne samplers is the most reliable method so far to precisely observe the mole fractions of atmospheric compositions from the tropopause to about 35 km. In the previous studies, air sampling has been carried out in the low latitude stratosphere (Volk et al., 1996; Patra et al., 1997; Schauffler et al., 1998; Andrews et al., 2001; Kaiser et al., 2006; Laube et al., 2010; Brinckmann et al., 2012). However, balloon measurements in the low latitudes, especially to high

altitudes (~30 km), have not been conducted as often as those in the middle and high latitudes, mainly due to the limited availability of balloon launching facilities.

To measure the $CO_2$ and $SF_6$ mole fractions, $\delta^{15}N$ of $N_2$, $\delta^{18}O$ of $O_2$, and $\delta(Ar/N_2)$ in the equatorial stratosphere, we conducted whole air sampling with balloon-borne cryogenic samplers over Biak, Indonesia in February 2015 as part of an observation campaign called "Coordinated Upper-troposphere-to-stratosphere Balloon Experiment in Biak" (CUBE/Biak)

(Hasebe et al., submitted to BAMS), in cooperation with the National Institute of Aeronautics and Space of the Republic of Indonesia (LAPAN). In this paper, we present measured vertical profiles of the $CO_2$ and $SF_6$ mole fractions, $\delta^{15}N$ of $N_2$, $\delta^{18}O$ of $O_2$, and $\delta(Ar/N_2)$, and discuss some of the implications of these measurements on the age and gravitational separation of stratospheric air. We also simulated the age and gravitational separation of air using a two-dimensional model (SOCRATES) (Huang et al., 1998; Park et al., 1999; Khosravi et al., 2002). Based on the simulation results, we discuss the effects of tropical upwelling and horizontal mixing on the age and gravitational separation of air.

## 2    Experimental Procedures

The collection of stratospheric air over Indonesia was performed using a balloon-borne cryogenic sampler. A map of the balloon launch site is shown in Figure 1. The cryogenic air sampler is equipped with a cooling device called "Joule–Thomson (J–T) mini cooler" in which neon is liquefied to be used as a refrigerant for collecting stratospheric air cryogenically (Morimoto et al., 2009). Eight sets of cryogenic air samplers and four large plastic balloons were prepared for the Biak campaign. Each balloon was loaded with two air samplers to collect stratospheric air at two different altitudes. The balloons were launched from the LAPAN observatory at Biak (001° 10'32" S 136° 06'02" E) on February 22, 24, 26, and 28, 2015; the samplers were recovered from a coastal area of Biak on the same day of each launch. Thus, we successfully collected stratospheric air samples at 8 altitudes from 17.2 to 28.7 km, with one at 17.2 and another at 18.5 km inside the Tropical Tropopause Layer (TTL). In the present study, air sampling was performed during balloon ascent. In the past, we conducted a number of similar air sampling using a cryogenic sampler with liquid helium (Honda et al., 1996) in which we collected samples during the balloon ascent and descent over Japan; they showed that the outgassing from the balloon and payload had negligibly small impact on the air sample quality even if air sampling was made during the balloon ascent (Morimoto et al., 2009; Nakazawa et al. 2002). One sampler assigned to 21 km leaked after landing due to a trouble in electric circuit. Overall, the amount of air collected ranged from 2.0 to 9.3 L at standard temperature (0°C) and pressure (1013.25 hPa), depending on the sampling altitude, enough to measure the mole fractions and isotopic ratios of various gases. The collected air samples were sent to Japan, and then analyzed for $\delta^{15}N$ of $N_2$, $\delta^{18}O$ of $O_2$, and $\delta(Ar/N_2)$ using a mass spectrometer and for $CO_2$ and $SF_6$ mole fractions using a non-dispersive infrared gas analyzer (NDIR) and a gas chromatograph (GC) equipped with an electron capture detector (ECD), respectively.

In this study, $\delta^{15}N$, $\delta^{18}O$, and $\delta(Ar/N_2)$ are defined as:

$$\delta^{15}N = \left( \frac{\left[ n\left(^{15}N^{14}N\right)/n\left(^{14}N^{14}N\right)\right]_{sample}}{\left[ n\left(^{15}N^{14}N\right)/n\left(^{14}N^{14}N\right)\right]_{standard}} -1 \right) \times 10^6 \quad (per\ meg), \tag{1a}$$

$$\delta^{18}O = \left( \frac{\left[ n\left(^{18}O^{16}O\right)/n\left(^{16}O^{16}O\right)\right]_{sample}}{\left[ n\left(^{18}O^{16}O\right)/n\left(^{16}O^{16}O\right)\right]_{standard}} -1 \right) \times 10^6 \quad (per\ meg), \tag{1b}$$

and

$$\delta(\text{Ar}/\text{N}_2) = \left( \frac{\left[ n\left(^{40}\text{Ar}\right)/n\left(^{14}\text{N}^{14}\text{N}\right) \right]_{\text{sample}}}{\left[ n\left(^{40}\text{Ar}\right)/n\left(^{14}\text{N}^{14}\text{N}\right) \right]_{\text{standard}}} - 1 \right) \times 10^6 \quad (\textit{per meg}), \tag{1c}$$

where $n$ means the amount of each substance, and "sample" and "standard" denote the sample and standard gases, respectively. Technical details of our mass spectrometry analyses have been described in Ishidoya and Murayama (2014). In this study, only the method of sample air flow into the mass spectrometer was modified from the previous method described in Ishidoya et al. (2013). With this modification, only a miniscule amount of sample air split off from an inlet system was transferred to the mass spectrometer through a fused silica capillary. While sample amount used for this method was larger than before, the precision was improved by one order of magnitude. In this study, we were able to determine $\delta^{15}$N, $\delta^{18}$O, and $\delta(\text{Ar}/\text{N}_2)$ more precisely than Ishidoya et al. (2013), with the respective reproducibility of about ±2, ±5, and ±7 per meg (±1σ), precise enough to detect small variations of interest in the stratosphere over the equatorial region. A detailed technical aspect of our $CO_2$ analysis has been given in Nakazawa et al. (1995) and Aoki et al. (2003). The $CO_2$ mole fraction was measured using an NDIR with analytical precision of better than 0.02 µmol mol$^{-1}$, employing our new $CO_2$ standard gases prepared in 2010 by a one-step dilution gravimetric method with an estimated uncertainty of 0.1 µmol mol$^{-1}$.

Since our $SF_6$ analysis procedure has not been published yet, a brief description is given here. The $SF_6$ mole fraction of each sample was determined twice at Tohoku University (TU) and twice at the Miyagi University of Education (MUE) against our $SF_6$ standard gas scale using their respective GCs. The analytical procedures at TU and MUE were basically the same, with a mixture of Ar (95%) and $CH_4$ (5%) used as a carrier gas for both GCs, but while TU employed Agilent 6890 with a packed column, MUE used 7890N with a capillary column. The volume of a sample loop was 10 mL for the TU's GC and 1 mL for the MUE's GC. Our $SF_6$ working standard gases used for the sample analysis were calibrated against our primary standard gases that were produced by the seven-step dilution gravimetric method. The relationships between the ECD signal and the mole fractions of the primary standard gases were approximated by quadratic equations. We prepared the primary standard gases (3, 5, 10, 30 pmol mol$^{-1}$, respectively) twice in 2001 (2001 scale) and 2002 (2002 scale), and found that the 2001 scale provides higher values by 0.10-0.15 pmol mol$^{-1}$ than the 2002 scale in a range of observed atmospheric $SF_6$ mole fractions. The 2001 scale was compared with those of other institutes in the 5th and 6th WMO/IAEA Round Robin Comparison Experiment programs (https://www.esrl.noaa.gov/gmd/ccgg/wmorr/), and the results showed that the 2001 scale is higher by 0.15 pmol mol$^{-1}$, on average, than the WMO X2006 scale. We also made an intercomparison of the $SF_6$ standard gas among TU, MUE, and the National Institute of Environmental Studies (NIES), Japan in August 2016. The results indicated that the 2001 scale is higher by 0.10-0.15 pmol mol$^{-1}$ than the NIES gravimetric scale, implying a good agreement between the 2002 scale and the NIES scale. Therefore, we decided to employ the 2002 scale for our measurements of the $SF_6$ mole fraction. As a result, our calibration scale agreed, to within our measurement precision, with WMO X2006 and NIES scales over the range of mole fractions observed in this study. The reproducibility of our $SF_6$ analysis was estimated to be better than 0.09 pmol mol$^{-1}$, based on one standard deviation (1σ) obtained from three replicate analyses of one air sample.

The $CH_4$ mole fraction of the air sample was also determined against our gravimetrically prepared $CH_4$ standard gas system using the GC with a flame ionization detector (Sugawara et al., 1997), since the measured value is necessary for the estimation of $CO_2$ age, as described below.

Possible deterioration of an air sample during storage should be closely examined and taken into account to accurately estimate the age of air from the measured mole fraction. From our experience of collecting air samples using the cryogenic sampler in Antarctica and Eastern Equatorial Pacific, we have found that the $CO_2$ mole fraction of air collected in stainless-steel bottle changes during sample storage, presumably due to $CO_2$ absorption on inner wall of the bottle. Although all the stainless-steel bottles used in this study were repeatedly evacuated prior to use, the $CO_2$ mole fraction still showed a slight decrease. Therefore, we carried out a sample storage test for each bottle to correct for the deterioration effect on $CO_2$ mole fraction. The correction amount ranged from 0.0 $\mu mol\ mol^{-1}$ to 0.7 $\mu mol\ mol^{-1}$, depending on the bottle. This deterioration effects have large influence on the age determination, because the correction of 0.7 $\mu mol\ mol^{-1}$ for $CO_2$ mole fraction is equivalent to 0.3 years of age correction. Therefore, the maximum age error was estimated by taking into account this deterioration for each sampler. We did not make a similar storage test for $SF_6$ prior to use; however, the $SF_6$ mole fraction of sample air was reanalyzed one month after the first set of analyses to check for possible changes in the mole fraction during storage period. Since changes in the $SF_6$ mole fraction were found to be 0.01-0.07 $pmol\ mol^{-1}$, the deterioration of $SF_6$ during the storage period was neglected. Change in the $CH_4$ mole fraction was also found to be within our measurement precision (Morimoto et al., 2009), and the impact of error propagation to the age determination was negligible.

## 3    Results and Discussion

### 3.1 Vertical profiles

Vertical profiles of the $CO_2$ and $SF_6$ mole fractions, $\delta^{15}N$ of $N_2$, $\delta^{18}O$ of $O_2$, and $\delta(Ar/N_2)$ in the TTL and stratosphere over Biak are shown in Figure 2. The exact values are given in Table 1 and 2. As shown in Figure 2, the $CO_2$ and $SF_6$ mole fractions are high in the TTL and decrease with increasing altitude. Considering that there are no sources and sinks of $CO_2$ and $SF_6$ in the stratosphere, except for a small amount of $CO_2$ production by $CH_4$ oxidation, the observed vertical profiles of the two variables are mainly formed by atmospheric transport processes. In particular, tropical upwelling in the atmosphere plays an important role in the mole fraction decrease with altitude, i.e., temporal variations of the mole fraction in the upper tropical troposphere are recorded as a vertical distribution due to the upward air transport from the TTL to the stratosphere. It is also seen from Figure 2 that the mole fractions do not change monotonically with altitude but decrease gradually from the TTL to 24 km and then rapidly to near constant values above 25 km. While the physical details of such complicated vertical $CO_2$ and $SF_6$ profiles are unclear, they can be reasonably reproduced by height-dependent upwelling and/or vertical and horizontal mixing. The vertical propagation of the $CO_2$ seasonal cycle from the troposphere will likely influence the vertical

distribution of the $CO_2$ mole fraction especially in the tropical lower stratosphere, since the seasonal amplitude in the tropical upper troposphere was observed to be larger than 3.3 μmol mol$^{-1}$ (as will be described later). In this connection, it is worth noting that the stratospheric water vapor observed during the Biak campaign period showed a clear tape recorder signal of similar behavior (Hasebe et al., submitted to BAMS). For a more quantitative study of the transport processes in the TTL and

tropical stratosphere, we need to take a multiple prong approach of integrating $CO_2$ and $SF_6$ data with other variables such as water vapor and $O_3$, using assimilated meteorological data and trajectory analyses. Similar $CO_2$ and $SF_6$ profiles have been reported by previous studies primarily in the northern mid-latitudes (Bischof et al., 1985; Schmidt and Khedim, 1991; Nakazawa et al., 1995, 2002; Aoki et al., 2003; Engel et al., 2009). However, the vertical gradient of the stratospheric $CO_2$ profile is much smaller in the tropics than in the mid-latitudes (e.g., Nakazawa et al., 1995; Aoki et al., 2003); a typical

difference of the $CO_2$ mole fraction between the lowermost and middle stratosphere (~30 km) over Japan is about 8 μmol mol$^{-1}$, whereas in the tropical stratosphere the difference is almost half of that observed in the mid-latitudes. We also measured vertical distributions of $CO_2$ and $SF_6$ from 15 to 35 km over Hokkaido, Japan (42° 30' N 143° 26' E) in August 2015 (Figure 2), using our traditional cryogenic sampler with liquid He (Nakazawa et al., 1995; Aoki et al., 2003). To compare the mole fractions of mid-stratospheric $CO_2$ and $SF_6$ in the tropics (Biak) with those observed in the northern

mid-latitudes (Hokkaido), average mole fractions were calculated at higher altitudes (potential temperatures larger than 600 K). The latitudinal differences in the $CO_2$ and $SF_6$ mole fractions were found to be 5.6 ± 0.9 μmol mol$^{-1}$ and 1.0 ± 0.2 pmol mol$^{-1}$, respectively. Considering that the air sampling over Hokkaido was carried out 6 months after Biak, these differences were obtained after taking into account the effects of the $CO_2$ and $SF_6$ increase and $CO_2$ production by $CH_4$ oxidation using the recent tropospheric increase rates of $CO_2$, $SF_6$ and $CH_4$. If the transit time of an atmospheric tracer from the tropics to

mid-latitudes can be simply obtained by dividing the latitudinal difference of the mole fraction between the two regions by its recent tropospheric increase rate, then the above observed differences would yield 2.4 ± 0.4 yrs for $CO_2$ and 3.1 ± 0.7 yrs for $SF_6$. These transit times correspond to aging of the air during the poleward transport from the equatorial region to mid-latitudes through the middle stratosphere. While the small difference in the transit time between $CO_2$ and $SF_6$ was not significant here, the difference in the mean age of air estimated from $CO_2$ and $SF_6$ will be discussed later.

As shown in Figure 2, $\delta^{15}N$ of $N_2$, $\delta^{18}O$ of $O_2$, and $\delta(Ar/N_2)$ decrease gradually with increasing altitude. Differences between the values at the lowermost and uppermost height levels (hereafter written as $\Delta\delta$) are 11.2, 18.4, and 153.3 per meg for $\delta^{15}N$, $\delta^{18}O$, and $\delta(Ar/N_2)$, respectively. Considering that the mass number differences of the related molecules are 1, 2, and 12 for $\delta^{15}N$, $\delta^{18}O$, and $\delta(Ar/N_2)$, respectively, $\Delta\delta$ seems to increase with increasing mass number difference. From a theoretical investigation of the molecular diffusion in polar firn air, the magnitude of the gravitational separation is proportional to mass

number difference (Etheridge et al., 1996), which can be expressed as,

$$\Delta\delta = \Delta m \times \Delta\delta_0. \tag{2}$$

Here, $\Delta m$ and $\Delta\delta_0$ are the mass number difference and the difference of δ values for $\Delta m=1$, respectively. Vertical distributions of the stable isotopic ratios of the major atmospheric components in the stratosphere were first reported by

Ishidoya et al. (2006, 2008a, 2008b). They concluded that the overall vertical structures of the isotopic ratios are basically caused by gravitational separation (Ishidoya et al., 2013), based on mass-dependent relationships between the related molecules. In this study, we also found, by examining $\Delta\delta$ in terms of the mass number differences of related molecules, that the relationship can be approximated by a proportional relationship, as shown in Figure 3. This suggests that the upward decreases of $\delta^{15}N$, $\delta^{18}O$, and $\delta(Ar/N_2)$ are mass dependent and that gravitational separation is observable in the stratosphere, not only in the mid-latitudes but also in the tropics. It is not clear what caused the small deviations of $\Delta\delta$ from the proportional relationship shown in Figure 3. The thermal diffusion is one of the plausible causes, but its effect on our observational data taken by using our traditional cryogenic sampler was negligibly small (Ishidoya et al., 2013).

After arranging all the observed $\delta^{15}N$, $\delta^{18}O$, and $\delta(Ar/N_2)$ measurements so that the values of the respective variables measured in the lowermost level become zero, we define an average value of $\delta^{15}N$, $\delta^{18}O$, and $\delta(Ar/N_2)$ as,

$$\langle\delta\rangle = \frac{1}{3}\left[\delta^{15}N + \delta^{18}O/2 + \delta\left(Ar/N_2\right)/12\right],$$ (3)

which is normalized for the mass number differences. Average vertical gradient of $\langle\delta\rangle$ above the tropopause was -3.3 ± 1.2 per meg km$^{-1}$ in the mid-latitude stratosphere (Ishidoya et al., 2013). On the other hand, our result shows that the average vertical gradient of $\langle\delta\rangle$ was only -1.4 ± 0.4 per meg km$^{-1}$ in the tropical stratosphere. The average $\langle\delta\rangle$ value at 14 hPa pressure level was about -35 per meg over Japan (Figure 1 in Ishidoya et al., 2013), while only -11 per meg in the tropical stratosphere at the same pressure level. This implies that the gravitational separation in the stratosphere is much weaker in the tropics than in the mid-latitudes.

### 3.2 Age of air

To estimate the age of stratospheric air using a clock tracer such as $CO_2$ or $SF_6$, a long-term record of its mole fraction observed in the upper troposphere is necessary as a reference. Since the tropospheric reference record is critical for calculating the stratospheric air age from the clock tracer data, it needs to be standardized. However, different data records have been used as a reference in different studies. For example, the data obtained at ground-based monitoring sites in tropical or subtropical regions were widely used for the tropospheric record after correcting for the mole fraction differences between the surface and the upper troposphere. Therefore, the estimated air ages have been plagued with large uncertainties due not only to the selection of the monitoring site, but also to the correction for the mole fraction difference between the site and the tropical upper troposphere. Uncertainties in the transport time from the boundary layer to the tropopause in the tropics have also been pointed out by Stiller et al. (2012). However recently, 10 or more-year mole fraction records of upper tropospheric $CO_2$ and $SF_6$ have become available from the Automatic Air Sampling Equipment (ASE) employed in the Comprehensive Observation Network for TRace gases by AIrLiner (CONTRAIL) program (Machida et al., 2008, Sawa et al., 2008; Matsueda et al., 2015). In this study, we have exploited the opportunity and constructed a set of tropical upper tropospheric $CO_2$ and $SF_6$ records from the CONTRAIL data obtained between 5° S and 5° N, between 142° E and 150° E

and between 10 and 13 km over Southeast Asia. The selected $CO_2$ and $SF_6$ data are shown in Figures 4 and 5, respectively, together with their best-fit curves obtained by using the Nakazawa et al. (1997) algorithm. Although we do not have sufficient knowledge about the temporal variations of their mole fractions near the top of the TTL to obtain a definitive age of the stratospheric air (Waugh and Hall, 2002), we decided to use the upper tropospheric CONTRAIL data records as a

reference. Therefore, the age of air estimated in this study could be slightly overestimated. Because the uncertainties associated with the best-fit curves will cause an error in the age estimation, we estimated the confidence intervals for the best-fit curves as the root mean squares of CONTRAIL data deviations from the curves, which were 0.65 µmol mol$^{-1}$ and 0.14 pmol mol$^{-1}$ for $CO_2$ and $SF_6$, respectively. These values were comparable or larger than the uncertainties of the mole fraction analyses. How the propagation of uncertainty impacts the age estimation will be discussed later.

The CONTRAIL $CO_2$ and $SF_6$ mole fractions are determined against the NIES scales, while the stratospheric data obtained in this study are expressed in the TU scales. However, as pointed out above, the differences in the $SF_6$ mole fraction scales between NIES and TU/MUE were found to be negligibly small. Therefore, our $SF_6$ data can be directly compared with the CONTRAIL $SF_6$ data. On the other hand, since we found that the TU $CO_2$ mole fraction scale gives slightly higher values than the NIES scale, the TU $CO_2$ data were shifted down by 0.20 µmol mol$^{-1}$, based on the results of the 5th and 6th Round

Robin inter-comparison programs (https://www.esrl.noaa.gov/gmd/ccgg/wmorr/wmorr_results.php?rr=rr6¶m=co2).
The age of stratospheric air at each altitude was estimated by comparing the corresponding $CO_2$ or $SF_6$ mole fraction with the related reference curve obtained from the CONTRAIL data. In the case of a linear time variation of the clock-tracer mole fraction, the mean age of air is determined by the difference in time between the observed mole fraction of a tracer in the stratosphere and the same value observed in the upper troposphere reference record. This is called the lag time (Waugh and

Hall, 2002). As seen in Figure 5, the $SF_6$ mole fraction shows no clear seasonal cycle in the tropical upper troposphere and its secular increase for the last 10 years can be approximated by a linear function. Therefore, the $SF_6$ age was sometimes reported as the lag time. However, the non-linear increase in the $SF_6$ mole fraction should be considered for a more precise mean age estimation. On the other hand, the tropospheric $CO_2$ mole fraction shows large seasonal cycle, as seen in Figure 4. Such a seasonal cycle propagates into the TTL and the lower stratosphere, and then gradually diminishes with increasing

altitude, due to air transport processes. Therefore, we cannot determine the $CO_2$ age from the lag time. The damping and phase delay of the seasonal $CO_2$ cycle in the stratosphere should be calculated theoretically using age spectra (Waugh and Hall, 2002). However, actual age spectra are usually unknown. Therefore, we used hypothetical age spectra to calculate the age of air. A simple formulation of the age spectrum was originally introduced by Hall and Plumb (1994) as a theoretical expression for a 1D atmosphere. Their proposed function is mathematically equivalent to the inverse-Gaussian distribution

(Waugh and Hall, 2002). We assumed that the width of the age spectrum ($d$) is parameterized using the mean age ($\Gamma$), i.e., $d^2$ = $0.7\Gamma$ (Waugh and Hall, 2002; Engel et al., 2008). In this study, the mole fraction of $CO_2$ or $SF_6$ in the stratosphere, $x(\Gamma, t)$, was calculated as a convolution of the tropospheric reference curve, $x_0(t)$, and the age spectrum, $G(\Gamma, t)$;

$$x(\Gamma,t) = \int_{t-\Delta t}^{t} x_0(t') \, G(\Gamma, t-t') dt', \tag{4}$$

where Δt is the integration time interval. Then, the $CO_2$ or $SF_6$ age was determined by comparing the observed mole fraction with x(Γ, t). As mentioned before, a small amount of $CO_2$ is produced by the chemical destruction of $CH_4$ in the stratosphere. To correct for this effect, each observed $CO_2$ mole fraction was adjusted using the $CH_4$ mole fraction measured from the same air sample prior to the age calculation. The corrections were less than 0.1 and 0.36 μmol mol$^{-1}$ for the lowermost and uppermost altitudes, respectively.

Ishidoya et al. (2013) pointed out that the mole fractions and isotopic ratios of all atmospheric constituents are influenced by gravitational separation and that its effect on a specified constituent can be evaluated from the <δ> value and the mass number difference between the constituent and the air. Since the molecular mass numbers of $CO_2$ and $SF_6$ are larger than the mean molecular mass of air, it is expected that their mole fractions are slightly separated gravitationally in the stratosphere, resulting in lower mole fractions compared to those in the absence of gravitational separation. In this study, we calculated the gravitational separation effect on the $CO_2$ and $SF_6$ mole fractions observed in the tropical stratosphere, and found that the respective effects are utmost −0.07 μmol mol$^{-1}$ and −0.01 pmol mol$^{-1}$, respectively. Therefore, the correction for gravitational separation is not necessary, at least up to ~30 km.

The $CO_2$ and $SF_6$ ages estimated in this study are shown in Figure 6, together with the <δ> values. These data are also tabulated in Tables 1 and 2. Considering the uncertainties associated with the $CO_2$ and $SF_6$ mole fraction measurements and the tropical tropospheric records, overall uncertainties of $CO_2$ and $SF_6$ ages were estimated by the following procedure. At first, normal pseudo random numbers multiplied by 1σ were added to the observed mole fraction data. The same procedure was applied to the tropical tropospheric records. Then, the age calculation procedure described above was repeated for 1000 different sets of random numbers, and the standard deviations of ages were calculated. The overall uncertainties in the estimated ages are also shown in Table 1. As shown in Figure 6, the $CO_2$ and $SF_6$ ages gradually increase with increasing altitude from the TTL to 22 km and then rapidly up to 25 km. The $CO_2$ and $SF_6$ ages show a good agreement with each other from the TTL to approximately 24 km, but they are significantly different above approximately 25 km. The average values of the $CO_2$ and $SF_6$ ages above 25 km showed a difference of about 1 year.

Our results show that the $CO_2$ and $SF_6$ ages were 0.5 - 0.6 yrs at the altitude of 18.5 km (~70 hPa), which roughly corresponds to the top of the TTL. In addition to this, the $CO_2$ and $SF_6$ ages increased slightly with increasing altitude inside the upper layer of the TTL. These results suggest that the $CO_2$ and $SF_6$ ages increased by 0.5 − 0.6 yrs from the tropical upper troposphere (approximately 11 ~ 13 km) to the top of the TTL. This indicates that a strict specification of the tropospheric reference record around the TTL is essential for the comparison between different observations. In this connection, it should be noted that our age estimates could be positively biased to some extent, since we used the CONTRAIL data as the tropospheric reference record.

Andrews et al. (2001) estimated the tropical lower stratospheric air age at 20 ± 0.5 km to be approximately 1.0 yrs, using the ER-2 aircraft. Note that their ages were estimated relative to the tropical tropopause. Even though it is not clear how different our age estimates are from the values estimated relative to the tropical tropopause, depending on the different

tropospheric reference records, our $CO_2$ and $SF_6$ ages at 22 km are nearly consistent with their results. The ages of air observed in the 25-30 km by balloon experiments were tabulated by Waugh and Hall (2002). Andrews et al. (2001) and Ray et al. (1999) estimated the $CO_2$ and $SF_6$ ages in the tropical mid-stratosphere (25–30 km) to be 3.5 and 4.0 yrs, respectively, from their balloon observations conducted at Juazeiro do Norte, Brazil (7° S) in 1997. Our results show that the $CO_2$ and $SF_6$ ages were 2.4 and 3.4 yrs on average at the altitudes of 25 - 29 km. If we compare our age data directly with those observed over Brazil, the $CO_2$ and $SF_6$ ages in this study are smaller by approximately 1.1 and 0.6 yrs, respectively. This would be partly due to the different time and observation location, although the balloon data observed in the equatorial mid-stratosphere are still relatively sparse and not representative in time and space.

The $SF_6$ age obtained in this study can be compared with the result obtained from the MIPAS satellite observations (Haenel et al., 2015). Haenel et al. (2015) estimated zonal mean $SF_6$ ages for each season by using the $SF_6$ mole fraction data from MIPAS for the period 2002-2012. They showed that the $SF_6$ age over the equatorial region in the northern winter season was approximately 1.0 yrs at 17 km and 5.0 yrs at 30 km, with the age increasing almost linearly with increasing altitude. Since the tropospheric reference record used for the MIPAS $SF_6$ age is ground-based global mean surface data, their age estimates are systematically larger than our estimates over the entire altitude range of the TTL to 30 km. By comparing our and the MIPAS $SF_6$ ages at 17 km, the difference in the two ages is found to be approximately 0.8 yrs. Taking this difference into account, the middle stratospheric $SF_6$ age obtained in this study is slightly lower than the MIPAS $SF_6$ age.

In general, the age values obtained in this study are smaller than those reported previously for the mid-latitude middle stratosphere (Engel et al., 2008; Ray et al., 2010; Ishidoya et al., 2013). Ishidoya et al. (2013) reported that the average value of $CO_2$ age at heights above 20 – 28 km over Japan was 5.2 ± 0.4 yrs for the period 1995-2010. However, their $CO_2$ age was estimated using ground-based data as the tropospheric reference record. Therefore, we re-evaluated their $CO_2$ age using the CONTRAIL data and found it to be 4.6 ± 0.4 yrs. Our latest observation over Japan in August 2015 showed the average middle stratospheric $CO_2$ age to be 4.9 ± 0.3 yrs (our unpublished data). The two values are also close to the $CO_2$ ages reported by Engel et al. (2008). From these results, we conclude that the middle stratospheric $CO_2$ age difference between the tropics and the mid-latitudes is approximately 2.2–2.5 yrs, which is consistent with the transit time roughly estimated from the $CO_2$ mole fraction in section 3.1. In summary, the $CO_2$ age increases vertically by approximately 3.0 yrs from the TTL to the middle stratosphere in the tropics and also horizontally in the middle stratosphere by approximately 2.0 yrs from the tropics to the mid-latitudes.

### 3.3 Difference in the $CO_2$ and $SF_6$ ages

The $SF_6$ age is obviously larger than the $CO_2$ age in the middle stratosphere, whether in the tropics or mid-latitudes. As mentioned before, this study shows that the average $CO_2$ and $SF_6$ ages above the height where the potential temperature was greater than 600 K over Biak are 2.4 ± 0.4 and 3.4 ± 0.5 yrs, respectively. On the other hand, our balloon observation over Japan in 2015 indicated an average $SF_6$ age of 6.8 ± 0.8 yrs for the middle stratosphere (potential temperatures larger than 600 K), which is 1.9 ± 0.9 yrs larger than the $CO_2$ age of 4.9 ± 0.3 yrs (our unpublished data). This result suggests that the

difference in the middle stratospheric $CO_2$ and $SF_6$ ages increases with latitude from the tropics (1.0 yrs) to the mid-latitudes (1.9 yrs).

The difference in the $CO_2$ and $SF_6$ ages has also been discussed in previous studies (Harnisch et al., 1998; Hall and Waugh, 1998; Strunk et al., 2000, Andrews et al., 2001). Harnisch et al. (1998) reported that the $CO_2$ age is smaller by up to 3 yrs

than the $SF_6$ age in the middle stratosphere, and then discussed the origin of excess $CO_2$. On the other hand, Strunk et al. (2000) showed from their balloon observation in 1997 that the $CO_2$ and $SF_6$ ages agree well with each other, and concluded that the discrepancy between the $CO_2$ and $SF_6$ ages reported by Harnisch et al. (1998) was caused by air sample degradation, such as $CO_2$ production during its storage in flasks. In this study, the sample degradation in the cryogenic sampler, as well as the mole fraction scales of $CO_2$ and $SF_6$ for stratospheric and tropospheric data, was carefully checked, as described in

section 2. Therefore, we were forced to find other causes for the difference between the $CO_2$ and $SF_6$ ages.

One possible cause is the propagation of the seasonal cycle of tropospheric $CO_2$ into the stratosphere. The propagation effect of periodic tracer variations into the stratosphere was also discussed in Andrews et al. (2001) and Waugh and Hall (2002). Since the seasonal cycle of the $CO_2$ mole fraction is significantly large in the tropical upper troposphere, its signal propagates into the TTL and then into the lower stratosphere via slow tropical upwelling. At the same time, the atmospheric

mixing process broadens the width of the age spectra, resulting in damped seasonal amplitude with increasing altitude. For an ideal clock tracer that has increased or decreased monotonically in troposphere, $x(\Gamma, t)$ will be a single-valued function of $\Gamma$, which allows us to determine the mean age of air from the clock tracer mole fraction. On the other hand, if the $CO_2$ seasonal cycle is still significantly large at the observation altitude, it is not necessarily guaranteed that $x(\Gamma, t)$ is a single-valued function of $\Gamma$, depending on the season. In such a case, the $CO_2$ age will be underestimated or overestimated,

depending on the time of year, and it is difficult to estimate the $CO_2$ age precisely from the mole fraction at that altitude. If that is the case, then the difference between the $CO_2$ and $SF_6$ ages caused by the $CO_2$ seasonal cycle might be significant in the season when the $CO_2$ mole fraction takes seasonal maxima and minima in the upper troposphere and the lower stratosphere. However, our results showed good agreement between the $CO_2$ and $SF_6$ ages in the TTL and the lower stratosphere. This is probably due to the fact that the seasonal $CO_2$ variation in the equatorial upper troposphere takes an

intermediate concentration value in February, a level between its maximum and minimum (Sawa et al., 2008). Therefore, it is likely that the propagation of the $CO_2$ seasonal cycle would be a minor contributor to the difference in the middle stratospheric $CO_2$ and $SF_6$ ages.

We also examined the influence of mesospheric air, because $SF_6$ is decomposed by UV absorption and electron reactions in the mesosphere (e.g., Reddmann et al., 2001). Andrews et al. (2001) reported that the $CO_2$ and $SF_6$ ages observed by ER-2

are in agreement with each other for ages less than 3.0 yrs, while the $SF_6$ age is 10–20% larger than the $CO_2$ age if older. They also suggested that the loss of mesospheric $SF_6$ would lead to an excess in age at higher altitudes. The age-dependent difference between the $CO_2$ and $SF_6$ ages obtained by Andrews et al. (2001) is nearly consistent with that observed in this study. However, it is not yet clear how much the mesospheric $SF_6$ loss affects the $SF_6$ age observed in the tropical middle

stratosphere.

The contribution of the mesospheric air subsidence to the stratospheric $SF_6$ age was also discussed by Stiller et al. (2012). They proposed that the "over-aging" of $SF_6$ in the mid- and high-latitude stratosphere could be due to the mixing of air with less $SF_6$ inside the polar vortex with the mid-latitude air, and estimated a rate of change in the northern hemispheric air age caused by the mixing to be 0.04 yrs per year of age. They also suggested that this effect is halved (0.02 yrs per year of age) if the mixing of the polar vortex air occurs throughout the stratosphere. Therefore, if the $SF_6$ age is estimated to be 5.0 yrs for a certain altitude in the northern hemisphere, the "true" $SF_6$ age should be 4.8 yrs or 4.9 yrs. More recently, Ray et al. (2017) also reported that the $SF_6$ age in the stratosphere must account for a potential influence from the polar vortex air. However, the difference between the $CO_2$ and $SF_6$ ages obtained in this study is so large that the "polar vortex mixing" effect is not a major factor. Linz et al. (2017) compared the MIPAS $SF_6$ age with the $N_2O$ age calculated with the $N_2O$ data from the Global OZone Chemistry And Related trace gas Data records for the Stratosphere (GOZCARDS), and showed that the MIPAS $SF_6$ age is larger than the $N_2O$ age in the tropics. The $CO_2$ and $SF_6$ ages observed in this study are consistent with the $N_2O$ age rather than the MIPAS $SF_6$ age, although the observation period is different.

The vertical distribution of gravitational separation, shown by $<\delta>$ in Figure 6, could give us useful information about possible process that produces the observed difference in the $CO_2$ and $SF_6$ ages in the middle stratosphere. As seen in Figure 6, the increases in the $SF_6$ age with increasing height in the upper layer are accompanied by the gravitational separation enhancement. Similar phenomena were also observed in the high latitudes from our previous balloon experiments (Ishidoya et al., submitted to ASL). Gravitational separation is completely independent of any chemical processes in the atmosphere, and therefore it could be an indicator of atmospheric transport processes (Ishidoya et al., 2013). Since gravitational separation will be highly enhanced in the upper stratosphere and the mesosphere, there is a possibility that the impact of $SF_6$ loss on the $SF_6$ age in the upper air or in the polar vortex can be evaluated by using gravitational separation data.

### 3.4 Numerical simulations of gravitational separation

The gravitational separation effect of atmospheric constituents is mainly governed by two factors: (1) mass-independent transport and (2) mass-dependent molecular diffusion; the former reduces the effect and the latter enhances it. In general, since the mass-dependent molecular diffusion flux of a specific constituent, $i$, (hereafter written as $F_{mi}$) is much smaller than the mass-independent flux, $F_{mi}$ is usually neglected except for the mass transport above the mesosphere. A theoretical expression for the vertical component of the molecular diffusion flux ($F_{mi,z}$) is given by:

$$F_{mi,z} = -D_{mi} \left\{ \frac{\partial n_i}{\partial z} + \frac{m_i g}{RT} n_i + \left(1 + \alpha_{Ti}\right) \frac{\partial \left(\ln T\right)}{\partial z} n_i \right\},$$

(5)

where $n_i$, $D_{mi}$, $m_i$, and $\alpha_{Ti}$ are the number density, the molecular diffusion coefficient, the molecular mass, and the thermal diffusion factor of a specific constituent, $i$, respectively, and $g$, $R$, and $T$ denote the gravitational acceleration, the gas constant and temperature, respectively (Banks and Kockarts, 1973). The relative difference in the number density ($n$) ratio of

constituent $i$ and $j$ between the stratosphere and the troposphere, $\delta_{i,j}$, is defined as:

$$\delta_{i,j} = \left( \left[ \frac{n_j}{n_i} \right]_{str} \middle/ \left[ \frac{n_j}{n_i} \right]_{trp} - 1 \right),$$

(6)

where subscripts $str$ and $trp$ denote the stratosphere and the troposphere, respectively. In the theoretical expression given by equation (5), the molecular dependence arises from $m_i$, $D_{mi}$, and $\alpha_{Ti}$. As described in section 3.1, $\Delta\delta$ is nearly proportional to

the mass number differences of the related molecules (see Figure 3), suggesting that the molecular mass difference is a main cause of molecular separation and $\alpha_{Ti}$ does not play an important role. Indeed, in a previous study (Ishidoya et al., 2013), we confirmed that the molecular separations of atmospheric major compositions due to the thermal diffusion are not proportional to the mass number differences. If $m_j$ is larger than $m_i$, the number density ratio, $n_j/n_i$, decreases with increasing altitude due to the mass-dependent flux, resulting in the upward decrease in the $\delta$ value (see Figure 6). In our observations of

the major atmospheric constituents, $i$ and $j$ correspond to $^{14}N^{14}N$ and $^{15}N^{14}N$, $^{16}O^{16}O$ and $^{18}O^{16}O$, or $N_2$ and $Ar$, for which $m_i < m_j$. Therefore, gravitational separation of two related molecules is enhanced as the $\delta$ value is lowered. In addition, the separation effect by molecular diffusion is enhanced with increasing altitude due to the rapid increase of the molecular diffusion coefficient, $D_{mi}$. Ishidoya et al. (2013) made a similar comment related to their results observed in the northern mid-latitude stratosphere. The most striking result obtained in this study is that the magnitude of gravitational separation is

significantly different between the mid-latitude and the equatorial regions. As described before, the magnitude of gravitational separation in the equatorial stratosphere is almost one third of that observed in the northern mid-latitude at 14 hPa. The TTL and the stratosphere over the tropics are characterized by a slow upward motion of air. We suggest that this tropical upwelling has a relatively large contribution to the mass-independent flux and significantly weakens the gravitational separation effect.

To examine the relative contributions of the tropical upwelling to the gravitational separation effect, we performed numerical simulations using a two-dimensional model of the middle atmosphere (SOCRATES) developed by the National Center for Atmospheric Research (NCAR) (Huang et al., 1998; Park et al., 1999; Khosravi et al., 2002). Since details of our model calculation have already been described in Ishidoya et al. (2013), a brief description is given here. In the present simulations, we neglected the thermal diffusion flux by setting $\alpha_{Ti}$, which appears in equation (5), to zero. Even by doing this, the term

involving the temperature gradient still remains in equation (5). This term does not represent the thermal diffusion flux, but is theoretically arisen from the flux components caused by concentration and pressure gradients (Banks and Kockarts, 1973). Then we calculated the $^{44}CO_2$ and $^{45}CO_2$ mole fractions to derive the $\langle\delta\rangle$ value. Note that the mass number difference between $^{44}CO_2$ and $^{45}CO_2$ molecules is 1. Therefore the model-calculated $\langle\delta\rangle$ value can be directly compared with the observed $\langle\delta\rangle$ value. Because our purpose is to simulate the gravitational separation in $^{44}CO_2$ and $^{45}CO_2$ molecules, it is not

necessary to reproduce $\delta(^{45}CO_2)$ variations in the actual atmosphere. Taking this into account, their mole fractions in the lowermost layer of the model were fixed at known values, assuming their mole fraction ratio, $n(^{45}CO_2)/n(^{44}CO_2)$, to be

constant. Before simulating the gravitational separation, we performed preliminary calculations to evaluate the time constant that characterizes how long it takes to attain a steady state. In the preliminary calculation, the initial $<\delta>$ value was set to zero in the entire atmosphere (i.e., molecules are not gravitationally separated), and then a 50-year calculation was performed for $^{44}CO_2$ and $^{45}CO_2$ mole fractions. From this calculation, the time constant of the gravitational separation was

estimated to be about 5 to 8 years at altitudes of 20–30 km. Considering this result, a 20-year spin-up calculation was carried out with no $^{44}CO_2$ and $^{45}CO_2$ increases in the lowermost layer, and then a 30-year simulation was performed in which $^{44}CO_2$ and $^{45}CO_2$ were monotonically increased at the model surface, without seasonal variations and keeping their mole fraction ratio constant. The age of air was also calculated from the model-simulated $CO_2$ mole fraction. The age of air was determined as the difference between the time when the same $CO_2$ mole fraction value was found in the tropical upper

troposphere and at a certain altitude in the stratosphere. Systematic difference between the calculated and observed ages was adjusted so that the age calculated for the TTL agrees with the observed value.

In our model simulations, we first calculated the $<\delta>$ value and the age of air under the SOCRATES baseline-atmospheric condition (Huang et al., 1998). However, the air age and $<\delta>$ calculated from this standard run were found to be significantly lower and higher, respectively, compared to the observed values. Previous studies also pointed out that the air age calculated

by the SOCRATES standard run is underestimated, probably due to a faster residual mean circulation (Park et al., 1999; Ishidoya et al., 2013). By examining the SOCRATES baseline condition, we found that the mean ascending rate at levels 18 to 24 km over the equator to be approximately 0.5 mm s$^{-1}$ (~16 km yr$^{-1}$). This value was obviously larger than the typical values of 0.2–0.3 mm s$^{-1}$ (6–10 km yr$^{-1}$) reported by previous studies (Mote et al., 1996; Randel et al., 2001). Therefore, we calculated the $<\delta>$ value and the age of air again by arbitrarily suppressing the residual mean meridional circulation to 60%

of the standard run (hereafter written as the "control run"), corresponding to a mean ascending rate of 0.3 mm s$^{-1}$ (~10 km yr$^{-1}$) over the equator. The average meridional distributions of the $<\delta>$ value calculated by the control run for DJF and JJA are shown in Figure 7, together with the results for the age of air. Ishidoya et al. (2013) reported that the average $CO_2$ age and $<\delta>$ in the 30-35 km height layer over Japan for JJA were 4.8 ± 0.4 years and -50 ± 19 per meg, respectively. Note that the $CO_2$ age in Ishidoya et al. (2013) was converted to the CONTRAIL-based value in this study. As shown in Figure 7 (c)

and (d), the calculated values of $<\delta>$ and the age of air at mid-stratosphere over northern mid-latitudes in JJA are nearly consistent with the results observed over Japan. However, the calculated age could not reproduce the rapid increase of $CO_2$ age with increasing height from tropopause to about 24 km and slower increase or almost constant $CO_2$ age above 24 km observed over Japan. The model result also shows that the vertical $<\delta>$ gradient is very small over the equator, rapidly increasing toward the pole. The vertical gradient of the age of air shows a similar variation (Figure 7 (b)). However, the

latitudinal difference in the vertical age gradient is smaller than that of the $<\delta>$ value. This phenomenon is caused by a strong dependence of the gravitational separation on the vertical air transport. In general, the gravitational separation is enhanced with increasing altitude due to the height dependency of the molecular diffusion coefficient. However, the upward mean flow from the lower part, where air compositions are not gravitationally separated, significantly weakens the molecular

separation effect at a certain altitude in the stratosphere.

To examine the equatorial $<\delta>$ dependency on the mean meridional circulation, average vertical profiles over the equator for DJF simulated by assuming different mean ascending rates, are shown in Figure 8, together with the observational results over Indonesia. As seen in this figure, the suppression of the mean meridional circulation improves the discrepancy between

the model-calculated and observed $<\delta>$ values over the equatorial region. However, the air age calculated by the control run is still underestimated, in particular above 24 km. To reduce the air age difference between the observation and simulation, we further suppressed the mean meridional circulation to 50% of the standard run. It is obvious from Figure 8 (b) that the age difference is reduced as expected, but the calculated age is still smaller than the observed value above 22 km. On the other hand, the further suppression of the mean meridional circulation yields an anomalous overestimation of gravitational

separation, implying that the vertical gradient of the $<\delta>$ value is sensitive to the ascending air motion. The overestimated effect of gravitational separation could arise from unrealistic mixing processes in the model atmosphere, such as horizontal and/or vertical eddy diffusion. To examine the contribution of horizontal mixing to the model-simulated air age and $<\delta>$ value, these two variables were re-calculated by intensifying the horizontal eddy diffusivity, $K_{yy}$, by a factor of 1.5. As shown in Figure 8, the result indicates that an upward increase in the air age is further enhanced due to additional aging by

the horizontal mixing. However, gravitational separation does not show any appreciable change, especially in the lower stratosphere, suggesting that gravitational separation is insensitive to horizontal mixing. From these results, it is suggested that the changes in the mean meridional circulation and the horizontal eddy diffusion have different impact on the gravitational separation and the age of air.

Our two-dimensional model results could not reproduce the observed vertical profiles of mean age of air and gravitational

separation by assuming a specific scenario, as shown in Figure 8. In order to extend our study on gravitational separation, a three-dimensional model study is needed. It is expected that a three-dimensional model incorporated with the molecular diffusion process can be constrained by gravitational separation data.

## 4    Conclusions

To investigate the age and gravitational separation of air in the equatorial stratosphere, we observed $CO_2$ and $SF_6$ mole

fractions, $\delta^{18}O$ of $O_2$, $\delta^{15}N$ of $N_2$, and $\delta$ ($Ar/N_2$) at altitudes of 17 to 29 km over Biak, Indonesia in February 2015, using a set of cryogenic samplers. The $CO_2$ and $SF_6$ ages were carefully estimated from their measured mole fractions by comparing with the CONTRAIL tropical upper tropospheric data, after applying necessary corrections to the mole fraction scales used in different laboratories. The $SF_6$ and $CO_2$ ages showed a general increase with height. The $SF_6$ age and its vertical gradient obtained in this study are nearly consistent with those obtained by MIPAS satellite observations. The $SF_6$ age is in good

agreement with the $CO_2$ age below 24 km, but increases at heights above 25 km, probably due to an over-aging of $SF_6$ resulting from the influence of upper air with low $SF_6$ mole fractions. To better understand the difference between the $SF_6$ and $CO_2$ ages above 25 km, it is necessary to investigate the effect of mesospheric $SF_6$ losses on the stratospheric $SF_6$ age in

detail. We also found a very small but significant upward decrease in <δ>. This implies that gravitational separation occurs within the entire stratosphere and its magnitude is altitude dependent. By comparing with the observational results at mid-latitudes, it was confirmed that the tropical upwelling weakens gravitational separation in the TTL and in the lower stratosphere. Simulations with the SOCRATES two-dimensional transport model revealed that gravitational separation is highly sensitive to the strength of tropical upwelling.

Uncertainties and issues raised in this study require further observations of the air compositions and model analyses for a better understanding of gravitational separation and age of air in the stratosphere. The knowledge obtained would be crucial for investigating stratospheric air transport processes. To enhance our understanding of the dynamics and chemistry in the TTL and the equatorial stratosphere in more detail, more comprehensive studies that include water vapor "tape recorder", other atmospheric constituents and their isotopic signatures are also necessary.

**Statement.** The authors declare that they have no conflict of interest.

**Acknowledgements.** We deeply thank the Scientific Ballooning (DAIKIKYU) Research and Operation Group of the Institute of Space and Astronautical Science (ISAS), JAXA, Japan and the staffs of LAPAN, Indonesia for their cooperation in our stratospheric air sampling. This study was supported by JSPS KAKENHI Grant Numbers JS26220101, JS15K05282 and 15H02814. This work was also selected and supported as the Small-Size Project by ISAS, JAXA. We also gratefully acknowledge the CONTRAIL team for providing their data for this study.

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

**Table 1. $CO_2$ and $SF_6$ mole fractions and ages observed over Biak, Indonesia in February 2015.**

| Date | Altitude (km) | $CO_2$ mole fractions ($\mu$mol mol$^{-1}$) | $SF_6$ mole fractions (pmol mol$^{-1}$) | $CO_2$-age (years) | $SF_6$-age (years) |
|---|---|---|---|---|---|
| Feb. 22, 2015 | 17.2 | 398.2 ±0.2 | 8.38 ±0.09 | 0.4 ±0.3 | 0.3 ±0.5 |
| Feb. 24, 2015 | 18.5 | 397.9 ±0.1 | 8.31 ±0.09 | 0.5 ±0.3 | 0.6 ±0.5 |
| Feb. 24, 2015 | 22.0 | 397.1 ±0.2 | 8.07 ±0.02 | 0.9 ±0.3 | 1.3 ±0.4 |
| Feb. 26, 2015 | 23.9 | 395.4 ±0.1 | 7.92 ±0.06 | 1.8 ±0.3 | 1.8 ±0.5 |
| Feb. 28, 2015 | 25.2 | 394.1 ±0.3 | 7.47 ±0.04 | 2.4 ±0.3 | 3.3 ±0.5 |
| Feb. 26, 2015 | 27.4 | 394.1 ±0.6 | 7.49 ±0.07 | 2.4 ±0.4 | 3.2 ±0.5 |
| Feb. 28, 2015 | 28.7 | 394.2 ±0.7 | 7.37 ±0.04 | 2.4 ±0.4 | 3.6 ±0.5 |

**Table 2. The values of $\delta^{15}N$ of $N_2$, $\delta^{18}O$ of $O_2$, $\delta(Ar/N_2)$, and $\langle\delta\rangle$ observed over Biak, Indonesia in February 2015. All the values are expressed as deviations from the measured values at 17.2 km.**

| Date | Altitude (km) | $\delta^{15}N$ of $N_2$ (per meg) | $\delta^{18}O$ of $O_2$ (per meg) | $\delta(Ar/N_2)$ (per meg) | $\langle\delta\rangle$ (per meg) |
|---|---|---|---|---|---|
| Feb. 22, 2015 | 17.2 | 0.0 | 0.0 | 0 | 0 |
| Feb. 24, 2015 | 18.5 | -0.4 | -2.4 | 10 | -0.3 ±1.0 |
| Feb. 24, 2015 | 22.0 | -1.9 | -1.0 | 1 | -0.8 ±1.0 |
| Feb. 26, 2015 | 23.9 | -3.2 | -5.4 | 16 | -1.6 ±2.5 |
| Feb. 28, 2015 | 25.2 | -3.8 | -9.1 | -43 | -4.0 ±0.5 |
| Feb. 26, 2015 | 27.4 | -4.6 | -14.0 | -44 | -5.1 ±1.7 |
| Feb. 28, 2015 | 28.7 | -11.2 | -18.4 | -153 | -11.1 ±1.8 |

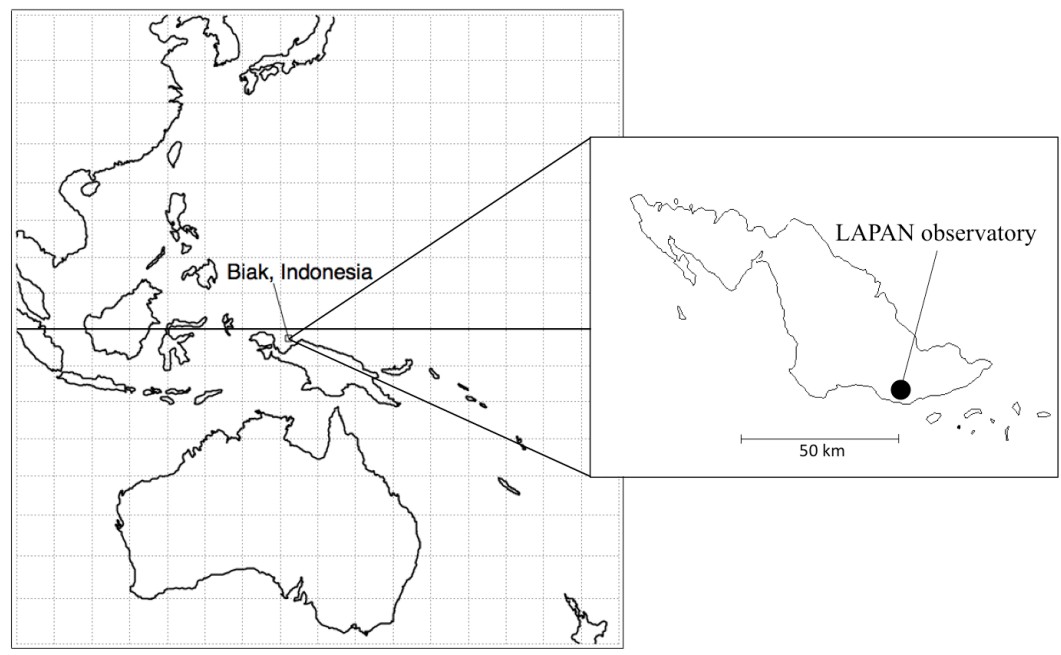

**Figure 1. Map showing Biak Island, Indonesia and the LAPAN observatory on the island.**

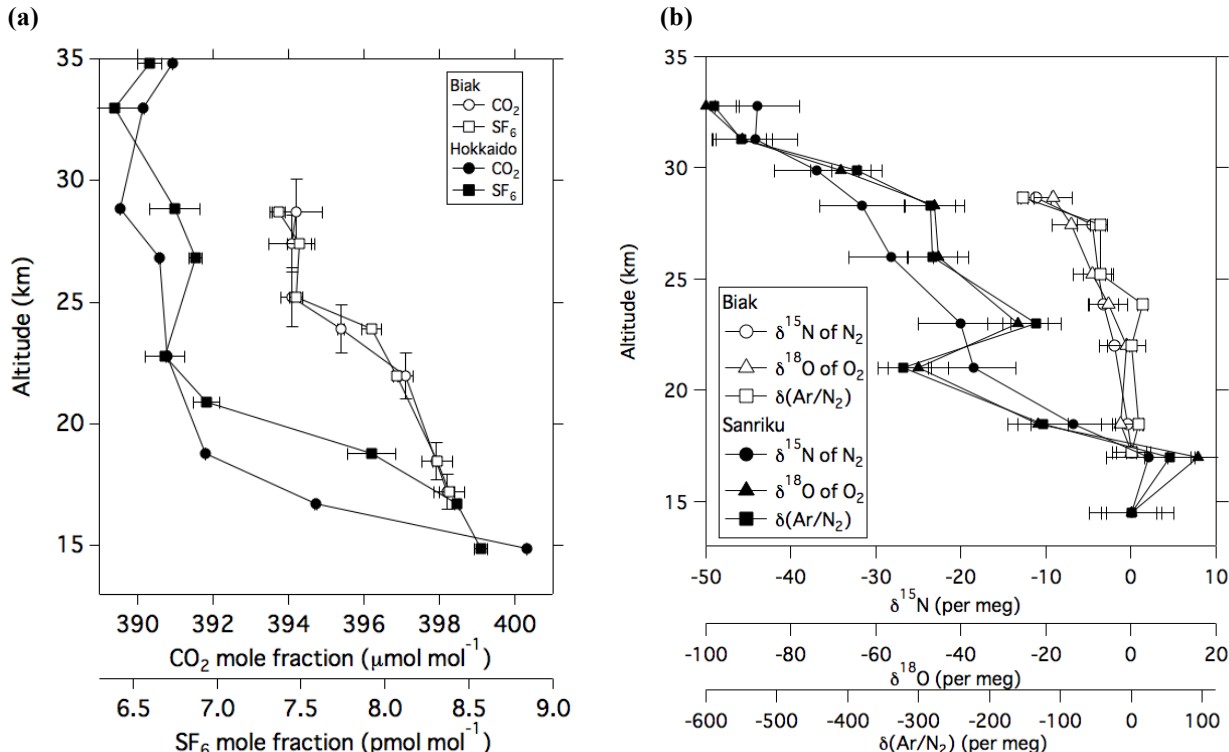

5    **Figure 2. (a) Vertical profiles of the $CO_2$ and $SF_6$ mole fractions observed over Biak, Indonesia on February 22–28, 2015, and Hokkaido, Japan (042° 30'00" N 143° 26'00" E) on August 6, 2015. Vertical error bars of $CO_2$ represent vertical ranges of air sampling. (b) Same as panel (a) but for $\delta^{15}N$ of $N_2$, $\delta^{18}O$ of $O_2$, and $\delta(Ar/N_2)$ observed over Biak and Sanriku, Japan (039° 09'39" N 141° 49'19" E) on June 4, 2007 (Ishidoya et al., 2013). $\delta^{15}N$ of $N_2$, $\delta^{18}O$ of $O_2$, and $\delta(Ar/N_2)$ observed over Sanriku are expressed as deviations from the measured values at 14.5 km.**

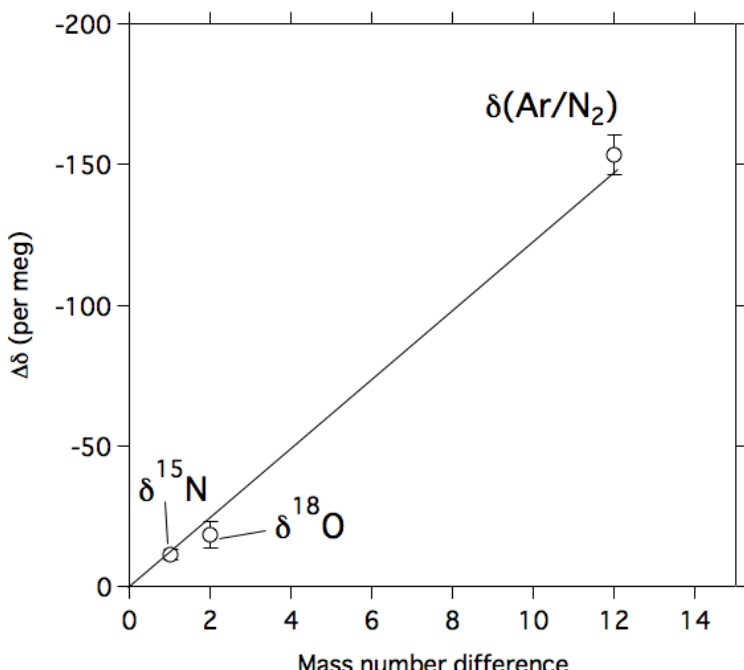

Figure 3. The relationship between the mass number difference and the $\Delta\delta$ value. The $\Delta\delta$ value denotes the difference in $\delta^{15}N$ of $N_2$, $\delta^{18}O$ of $O_2$ or $\delta(Ar/N_2)$ between 17.2 and 28.7 km. Solid line represents a linear relationship between $\Delta\delta$ and the mass number difference.

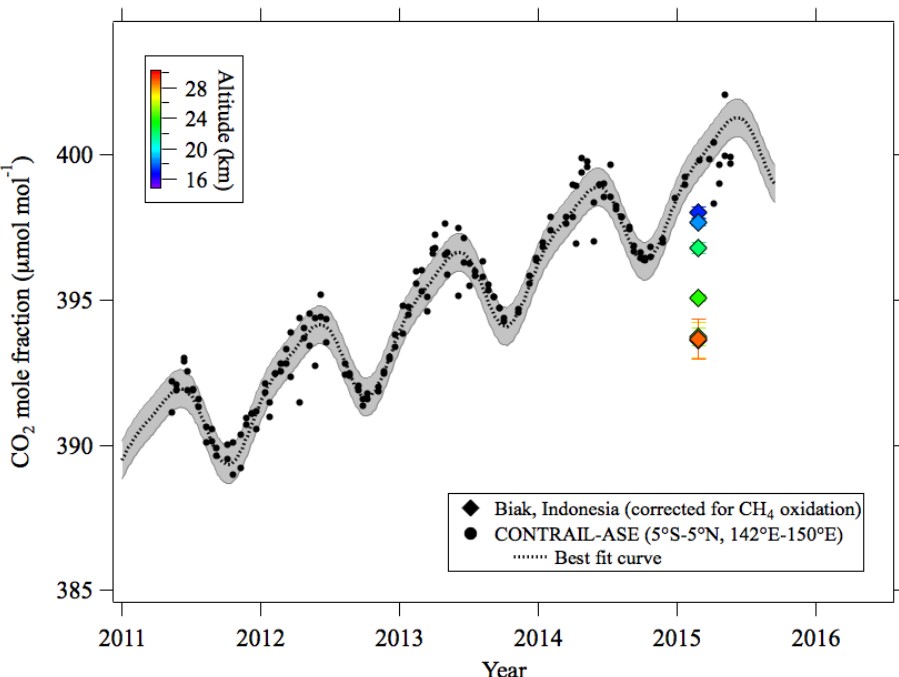

5    Figure 4. Stratospheric CO$_2$ mole fraction values observed over Biak and tropical upper tropospheric data (5° N–5° S, 142° E–150° E, 10–12.5 km) obtained by the CONTRAIL aircraft project. Also shown is the best-fit curve fitted to the CONTRAIL data. Gray shade shows one standard deviation width calculated by the fitting procedure.

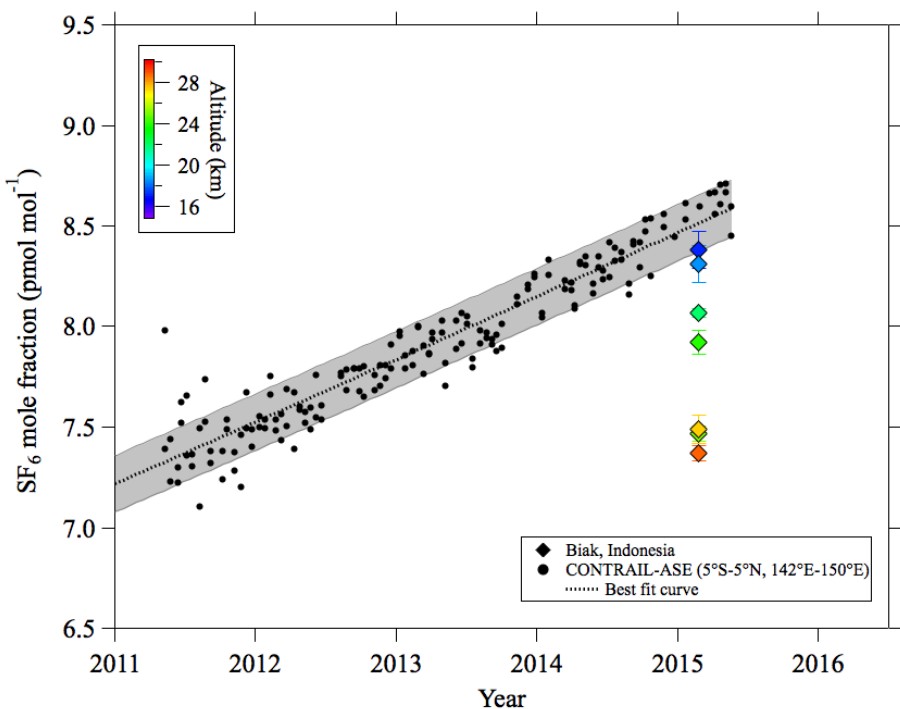

**Figure 5. Same as Figure 4 but for SF$_6$.**

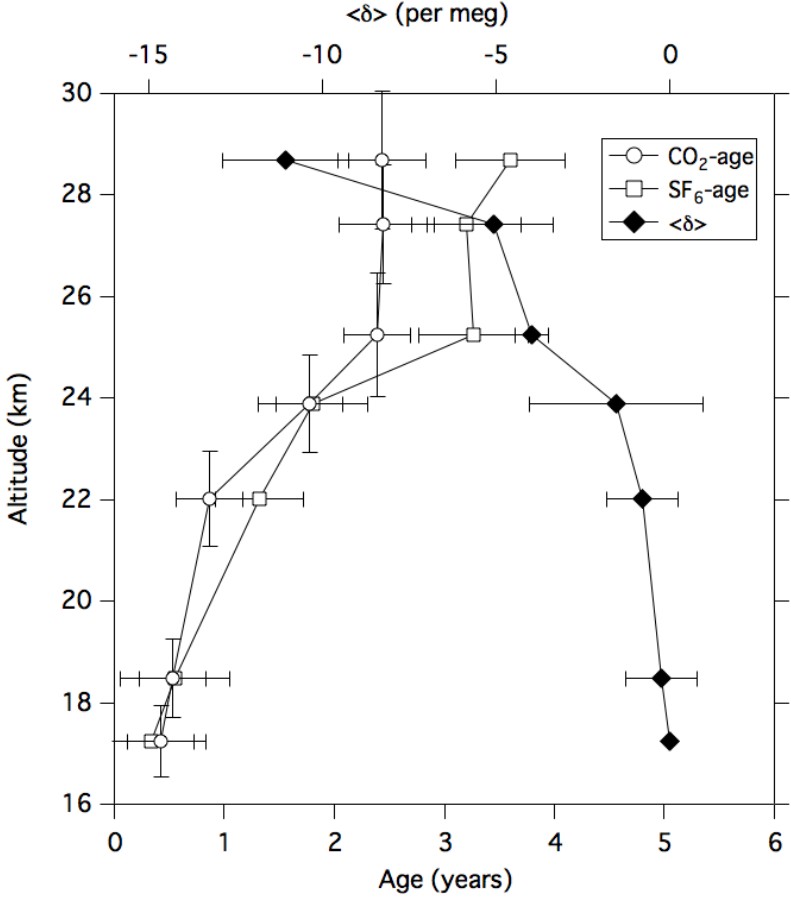

**Figure 6. Vertical profiles of the $CO_2$ and $SF_6$ ages and the $<\delta>$ value over Biak, Indonesia on February 22–28, 2015. The $<\delta>$ value represents an average of $\delta^{15}N$ of $N_2$, $(\delta^{18}O$ of $O_2)/2$, and $\delta(Ar/N_2)/12$ (cf. text). Vertical error bars of $CO_2$ age represent vertical ranges of air sampling.**

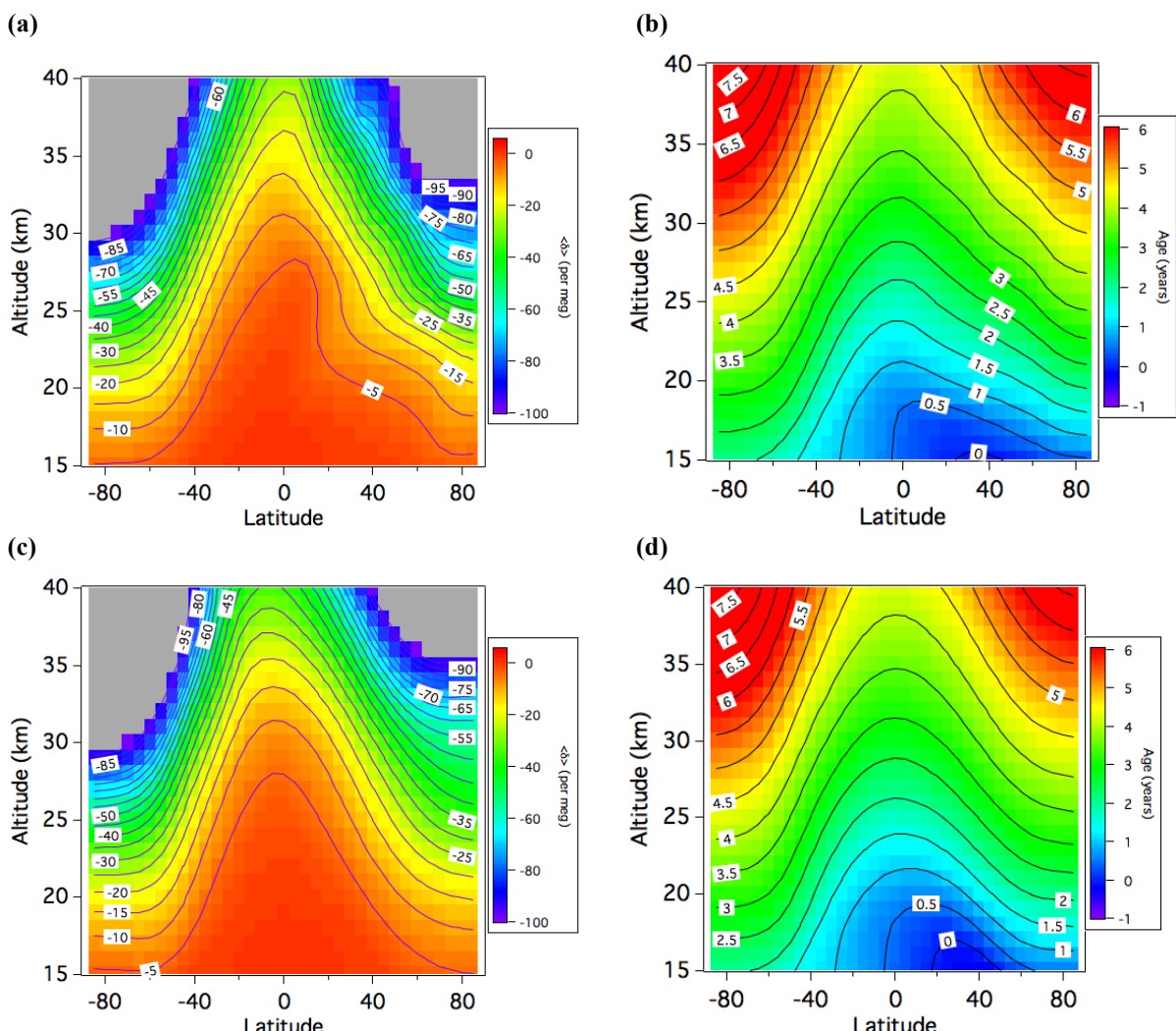

Figure 7. (a) Average meridional distributions of the <δ> value for DJF calculated using the SOCRATES model with the control run (cf. text). The <δ> values lower than −100 per meg are shown in gray. (b) Same with (a), but for the age of air. (c, d) Same as (a) and (b), respectively, but for JJA.

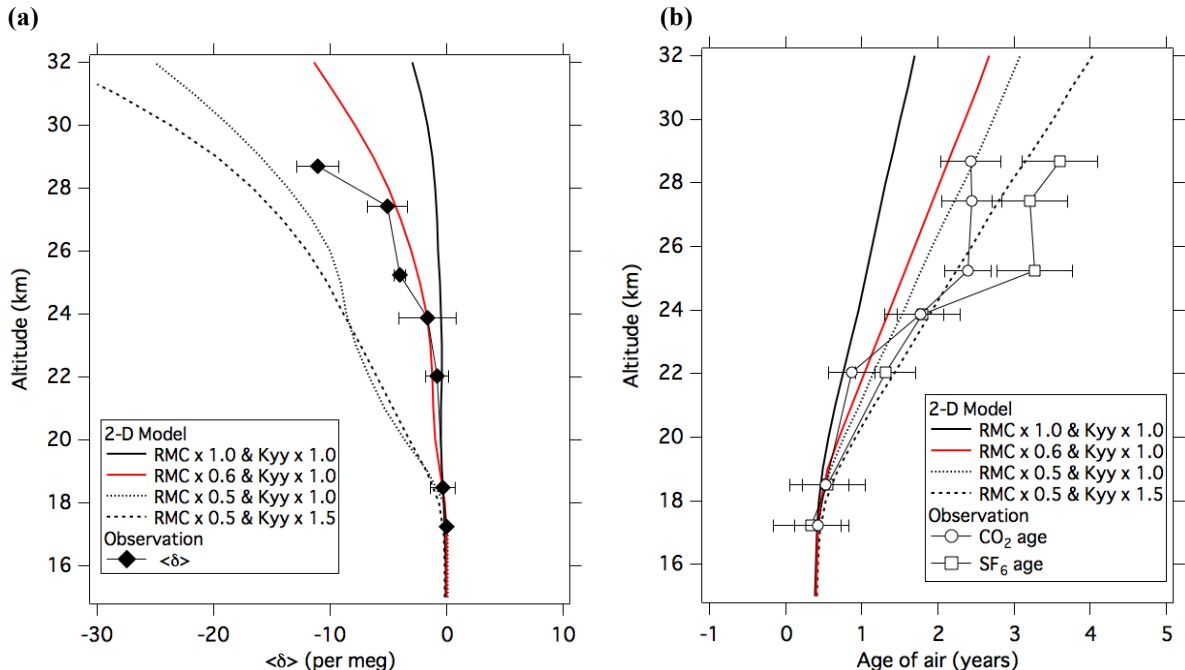

**(a)**

**(b)**

Figure 8. (a) Average vertical profiles of <δ> over the equator for DJF, calculated using the SOCRATES model as a standard run (black solid line), suppressed residual mean circulation (RMC) runs (red line and black dotted line), and suppressed residual mean circulation and enhanced horizontal eddy diffusion ($K_{yy}$) run (black dashed line). The <δ> values observed over Biak are also plotted (closed diamonds). (b) Same as in panel (a) but for the age of air. The observed $CO_2$ and $SF_6$ ages are shown by open circles and squares, respectively. Model results are shifted so that the age values at 17 km are equal to $CO_2$ age observed at 17.2 km.