# Peer review of "Age and gravitational separation of the stratospheric air over Indonesia"

_Atmospheric Chemistry and Physics, 2017_

## Referee Comment (RC1) · Anonymous Referee #1 · 7 Sep 2017

This manuscript reports a small number of observations of two trace gases, CO2 and SF6, as well as the isotopic composition of N2 and O2, and the ratio of Argon over the former. These observations were carried out with an innovative technique in an interesting and understudied region, i.e. the tropical stratosphere and the authors report some interesting effects. I recommend the manuscript for publication in ACP in general, but a large number of issues will need addressing first; the main ones being

1) a better placing of the results in context of existing literature and comparisons with published results, in particular for the mean age of the air.

2) clarification of many questions related to sample collection, measurements, error bar calculation, and quality assurance.

3) an improved discussion of results relative to common vertical coordinates in the stratosphere, such as the altitude relative to the tropopause, pressures or potential temperatures.

4) a wider consideration of potential age uncertainties and bias and reassessment of the difference between $CO_2$ and $SF_6$-derived mean aged and whether or not it is indeed significant.

5) the modelling part of the manuscript is a nice addition, but currently rather inconclusive as none of the tested scenarios seems to be able to match the observations.

More specific points can be found below.

P1, l15 It is common to open with a statement on the motivation for and wider context of the work presented. I encourage the authors to adopt this practice as it also tends to improve readability.

P1, l17 The concept of the mean age of air needs some introduction here.

P1, l20 Should this be 29 km instead of 25?

P1, l30 Again, the concept of the age of air is not introduced at all.

P2, l1 There is also the more recent work of Engel et al., ACP, 2017 extending the balloon-based trend as well as Stiller et al., ACP, 2017 who show that the observed latitudinal and vertical patterns of the decadal changes of age of air in the lower to middle stratosphere during 2002–2012 are predominantly caused by a southward shift of the circulation pattern of about 5 degrees

P2, l2 Only in the region of constant mean age above 25 km. In fact, both Diallo et al., ACP, 2012 and Boenisch et al., ACP, 2011 found indications for changes in the region below 25 km.

P2, l8-10 Ploeger et al., 2015 demonstrated that both mixing and residual circulation impact on ageing throughout the stratosphere.

P2, l20-22 This is not correct. There is substantially more literature on air sampling in those regions: Volk et al., 1996, Schauffler et al., 1998, Tuck et al., 2004, Kaiser et al., 2006, Laube et al., 2010, and Brinckmann et al., 2012 to name but a few. It would be good to see how the results, in particular the mean ages, compare.

P3, l1-5 Were these samples collected during ascent or descent? In either case, can the authors provide evidence for all species measured that no contamination originated from the balloon (in particular if collection occurred during ascent – Helium leaking into samplers is a known problem) or from the payload?

P3, l20 What was the reason for the improved precision?

P4, l6-7 What range of mole fractions was covered by those primary standards? ECDs are known to be strongly nonlinear detectors, so what methods were employed to ensure that a) detector responses were corrected and b) there is agreement between calibration scales over the entire range of observed mole fractions?

P4, l25 If $CO_2$ mole fractions do indeed change over time, the exact change will be quite influential on the determined age of air. 0.7 $\mu$mol mol$-1$ is equivalent to several months of mean age and it would be good to know whether the authors have tested any potential variability of such a correction from repeats of storage experiments as well as different water vapour contents, both of which are known to be influential factors. A similar question arises for methane – has this been storage-tested at all?

P5, l4-5 This is interesting and unusual as the upper part of the profile implies a vertical transport barrier in the tropical stratosphere. Has this been observed in other tropical data? Could this be a regional phenomenon or related to the onset of the recent QBO anomaly? Or is there any evidence for impact from convection? Moreover, can the authors explain why $CO_2$ and $SF_6$ should exhibit a similar behaviour in the lower part of the profile when one has a strong seasonal cycle and the other has none? $CO_2$ would surely be expected to show some variant of a tape-recorder signal, unless the authors have evidence to the contrary?

P5, l13-16 It is not clear what is being compared here. Similar altitudes cannot be compared directly between tropical and mid latitudes as tropopause heights differ substantially. The authors could use alternative coordinates such as height above tropopause, potential temperature or potential vorticity to address this problem. This would also help answering the question of where the TTL actually was during sampling. In addition, any comparison requires the inclusion of the mid-latitude data set, which could for instance be added to Figure 2.

P6, l9-12 This is not a valid comparison. In the mid-latitudes, 32 km could mean well over 20 km into the stratosphere, whereas in the tropics the tropopause could be around 19 km. Gravitational separation should be compared at least on a per km in the stratosphere basis and the effects of comparing different pressures should also be considered.

P6, l27-28 Why are only the last 5 years shown of 10 or more are available? Are the fits less good for the earlier years? What is the uncertainty range of those fits and how high are the resulting uncertainties in mean ages? And why does the data in Figures 4 and 5 have no error bars?

P7, l11-12 According to the major ground-based networks (AGAGE and NOAA) global SF6 mole fractions did not increase linearly in the last decade. Emissions have been increasing (WMO, 2014) resulting in some curvature of that trend. Have the authors considered the potential impact on mean ages, or, if they cannot resolve these curvatures, the additional uncertainty in their mean ages?

P8, l8 How were these uncertainties calculated?

P8, l9-11 Do the authors actually have any data (e.g. temperature or pressure) to confirm where the tropopause exactly was? And what is meant by "secondary tropopause"?

P8, l24-25 Can the authors present any evidence for this claim?

P8, l29 So other observations do not show a step change in age around 24 km. This reemphasizes my earlier question on an explanation for that phenomenon.

P9, l6 Again, not necessarily relative to the tropopause.

P9, l17-24 There are several more important and also more recent studies, e.g. Waugh and Hall, 2002 and in particular Ray et al., 2017.

P10, l19-20 I suggest showing the actual correlation in Figure 6. I would certainly say the two quantities do not correlate well around the upper part of the profile.

P10, l24-25 This is quite a claim. Are the authors suggesting that the tropical pipe does not pose a significant transport barrier between tropics and extra-tropics?

P11, l6-10 It certainly suggests that the molecular mass difference is dominant, but the molecular diffusion flux could still dampen the signal considerably. A mere proportionality does not rule that out.

P11, l16-17 As pointed out earlier the differences in pressures and temperatures at similar altitudes are quite important here and should be discussed.

P12, l8-9 Did the model-simulated $CO_2$ mole fraction include a realistic seasonal cycle?

P12, l23-25 Figure 7 does not include a comparison with data from Ishidoya et al., 2013. I suggest either including it in the figure or as quantitative statements in the text.

P21&26, figures 2 and 6 Surely the air samples were collected over a range of altitudes? Please add the y-axis uncertainties for all relevant figures.

P23&24, figures 4&5 Some of the scatter observed here is very likely due to changing input from the northern and southern hemisphere as the ITCZ moves through that latitudinal band. Can the authors assess the potential age uncertainty resulting for the exact time of their balloon campaign? This effect might even introduce a bias.
* * *
[Figure]

2017.

---

## Referee Comment (RC2) · Anonymous Referee #2 · 12 Sep 2017

This paper extends the investigation in earlier papers, by several of the same authors, of the gravitational separation of various species in the stratosphere, and its possible implications for atmospheric transport. What is novel, and welcome, about this paper is its focus on tropical data. In principle, the paper appears to be worthy of publication, subject to attention to a few issues, outlined in what follows.

Gravitational separation does indeed, as the authors claim, provide a new perspective on stratospheric transport, but it is not made very clear just what that perspective is. Some "ad hoc" experiments are illustrated in the 2D model, in which transport parameters are changed, but things would be made much more clear if there were a theoretical exposition of the problem. For example, one could use simplified models to show how separation would manifest itself in the presence of upwelling alone, or verti-

cal eddy diffusion alone. These would not reproduce the real world, but would provide some theoretical baseline to strengthen understanding of what a more complete model shows. One further shortcoming of the model perturbation experiments discussed in section 3.4 is that the tropics are considered in isolation from the rest of the globe. The authors may get better fits with the tropical data by changing parameters (upwelling, mixing) but that could be at the expense of agreement in the extratropics.

The relationship with the vast literature on stratospheric transport could be better illustrated by citing some of this literature more extensively than has been done. Some suggestions are outlined in the following.

Other issues, as they arise in the text (page, line):

(1,32): Determination of the BD circulation from age observations has been explicitly discussed in a recent paper by Linz et al. (Nature Geosci., 2017).

(2,10): The separate effects of circulation and mixing on age distributions are discussed in Garny et al. (J. Geophys. Res., 2014) and Linz et al. (J. Atmos.Sci., 2016). The strength of the circulation determines the horizontal gradient of age, rather than age itself; both mixing and circulation determine the vertical structure.

(5,31): Is it obvious that gravitational separation depends on mass number difference, rather than, say, mass ratio? If this is a theoretical prediction, please describe it.

(6,3): The claim that Figure 3 suggests a linear relationship seems a little exaggerated. There are basically only two points (to be sure, there are 3, but two of them are very close together). Empirically, one could fit any number of curves to the data shown. If there is an a priori expectation of a straight line (from theory, or from more extensive observations) then you should say so. Further, the line does not appear to pass through the origin, which surely it should?

(8, 25) and elsewhere: Nowhere is it acknowledged that the authors are trying to draw conclusions from data that are highly localized in time, and whose representativeness

is therefore open to question. So, e.g., the differences with the Brazil data may due to temporal, rather than spatial, variations. In general, the limitations of the temporal sampling should be prominently acknowledged.

(9,18): The differences between $CO_2$- and $SF_6$-based age calculations are also discussed in, e.g., Hall and Waugh (J. Geophys. Res., 1998), Ray et al. (J. Geophys. Res., 2017), Linz et al. (Nature Geosci., 2017).

(10,1): Since the $CO_2$ seasonal cycle, like the water vapor "tape recorder" signal, propagates into the tropical lower stratosphere as a decaying sinusoid in the vertical, I do not understand why "the age difference should be larger in the lower stratosphere . . .". It depends on time of year of the data being compared.

(11,15): If it is the mass dependence of $D_{mi}$ that matters most, could you show us what that dependence is?

(11,25): Given the relatively small variation of T in the stratosphere, does this term matter much in practice?

Figure 7: Why are the altitude scales different on the two frames? And can you comment on the negative ages in the northern lower stratosphere?

---

## Author Comment (AC1) · 25 Oct 2017

Reply to the referee #1

We would like to thank the referee for the constructive comments. We have tried to address all of them as detailed below. Our point-by-point response is typed in *italics: referee's comments*, roman: authors' response. Text in the revised manuscript is shown in red with page and line numbers of the new revised manuscript in squared brackets.

*This manuscript reports a small number of observations of two trace gases, CO2 and SF6, as well as the isotopic composition of N2 and O2, and the ratio of Argon over the former. These observations were carried out with an innovative technique in an interesting and understudied region, i.e. the tropical stratosphere and the authors report some interesting effects. I recommend the manuscript for publication in ACP in general, but a large number of issues will need addressing first; the main ones being*
*1) a better placing of the results in context of existing literature and comparisons with published results, in particular for the mean age of the air.*

We have added many reference articles mainly for the mean age studies, and revised or added some sentences describing the mean age comparisons, as follows.

[p2, L3-5] Balloon and satellite observations (Engel et al., 2009, 2017; Stiller et al., 2012) found that the age of air derived from $CO_2$ and $SF_6$ in the stratosphere over the northern mid-latitudes did not show any significant trend above 25 km over the last 30 years,

[p2, L9-14] Bönisch et al. (2011) suggested that the increased upwelling in the tropics after 2000 enhanced the lower stratospheric transport from the tropics into the extra-tropics. From an analysis of the ERA-Interim dataset, Diallo et al. (2012) also showed a negative trend over the 1989–2010 period in the lower stratosphere below 25 km. Linz et al. (2017) discussed the strength of the meridional overturning circulation of the stratosphere by using satellite observation data of $SF_6$ and $N_2O$, and suggested that a mesospheric $SF_6$ loss is important for age estimation using $SF_6$ mole fraction in the upper layer.

[p2, L17-19] Therefore, to discuss a change in the mean age estimated using the clock tracer, it is important to separately evaluate the respective effects of mean circulation and mixing processes on the air age (Garny et al., 2014; Ploeger et al., 2015; Linz et al., 2016).

[p2, L27-29] Air sampling has been carried out in the low latitude stratosphere (Volk et al., 1996; Patra et al., 1997; Schauffler et al., 1998; Andrews et al., 2001; Kaiser et al., 2006; Laube et al., 2010; Brinckmann et al., 2012).

[p10, L16-17] Taking this difference into account, the middle stratospheric $SF_6$ age obtained in this study is slightly lower than the MIPAS $SF_6$ age.

[p11,L4-5] The difference in the $CO_2$ and $SF_6$ ages has also been discussed in previous studies (Harnisch et al., 1998; Hall and Waugh, 1998; Strunk et al., 2000, Andrews et al., 2001).

[p12, L8-9] More recently, Ray et al. (2017) also reported that the $SF_6$ age in the stratosphere must account for a potential influence from the polar vortex air.

[p12, L11-14] Linz et al. (2017) compared the MIPAS $SF_6$ age with the $N_2O$ age calculated with the $N_2O$ data from the Global OZone Chemistry And Related trace gas Data records for the Stratosphere (GOZCARDS), and showed that the MIPAS $SF_6$ age is larger than the $N_2O$ age in the tropics. The $CO_2$ and $SF_6$ ages observed in this study are consistent with the $N_2O$ age rather than the MIPAS $SF_6$ age, although the observation period is different.

*2) clarification of many questions related to sample collection, measurements, error bar calculation, and quality assurance.*

In the section of experimental procedures, we revised the sentences to describe the balloon measurement, sample air analyses, and sample air quality in more detail. Possible deterioration effect on air sample is essential for our age determinations. We have reassessed uncertainties of our measurements. Uncertainties shown in Table 1 and error bars for the observed values in Figure 2, 4, and 5 have been also revised or added.

[p3, L16-20] In the present study, air sampling was performed during balloon ascent. In the past, we conducted a number of similar air sampling using a cryogenic sampler with liquid helium (Honda et al., 1996) in which we collected samples during the balloon ascent and descent over Japan; they showed that the outgassing from the balloon and payload had negligibly small impact on the air sample quality even if air sampling was made during the balloon ascent (Morimoto et al., 2009; Nakazawa et al. 2002).

[p4, L4-8] In this study, only the method of sample air flow into the mass spectrometer was modified from the previous method described in Ishidoya et al. (2013). With this modification, only a miniscule amount of sample air split off from an inlet system was transferred to the mass spectrometer through a fused silica capillary. While sample amount used for this method was larger than before, the precision was improved by one order of magnitude.

[p4, L20-24] We prepared the primary standard gases (3, 5, 10, 30 pmol mol$^{-1}$, respectively) twice in 2001 (2001 scale) and 2002 (2002 scale), and found that the 2001 scale provides higher values by 0.10-0.15 pmol mol$^{-1}$ than the 2002 scale in a range of observed atmospheric $SF_6$ mole fractions. The relationships between the ECD signal and the mole fractions of the primary standard gases were approximated by quadratic equations.

[p4, L30-31] As a result, our calibration scale agreed, to within our measurement precision, with WMO X2006 and NIES scales over the range of mole fractions observed in this study.

[p5, L9-18] Therefore, we carried out a sample storage test for each bottle to correct for the deterioration effect on $CO_2$ mole fraction. The correction amount ranged from 0.0 μmol mol$^{-1}$ to 0.7 μmol mol$^{-1}$, depending on the bottle. This deterioration effects have large influence on the age determination, because the correction of 0.7 μmol mol$^{-1}$ for $CO_2$ mole fraction is equivalent to 0.3 years of age correction. Therefore, the maximum age error was estimated by taking into account this deterioration for each sampler. We did not make a similar storage test for $SF_6$ prior to use; however, the $SF_6$ mole fraction of sample air was reanalyzed one month after the first set of analyses to check for possible changes in the mole fraction during storage period. Since changes in the $SF_6$ mole fraction were found to be 0.01-0.07 pmol mol$^{-1}$, the deterioration of $SF_6$ during the storage period was neglected. Change in the $CH_4$ mole fraction was also found to be within our measurement precision (Morimoto et al., 2009), and the impact of error propagation to the age determination was negligible.

*3) an improved discussion of results relative to common vertical coordinates in the stratosphere, such as the altitude relative to the tropopause, pressures or potential temperatures.*

We thank the referee for this critical comment. We have added or revised related discussions by using pressure and potential temperature especially for the comparisons between tropics and mid-latitudes, as follows.

[p6, L13-17] To compare the mole fractions of mid-stratospheric $CO_2$ and $SF_6$ in the tropics (Biak) with those observed in the northern mid-latitudes (Hokkaido), average mole fractions were calculated at higher altitudes (potential temperatures larger than 600 K). The latitudinal differences in the $CO_2$ and $SF_6$ mole fractions were found to be 5.6 ± 0.9 μmol mol$^{-1}$ and 1.0 ± 0.2 pmol mol$^{-1}$, respectively.

[p7, L13-17] Average vertical gradient of $<\delta>$ above the tropopause was -3.3 ± 1.2 per meg km$^{-1}$ in the mid-latitude stratosphere (Ishidoya et al., 2013). On the other hand, our result shows that the average vertical gradient of $<\delta>$ was only -1.4 ± 0.4 per meg km$^{-1}$ in the tropical stratosphere. The average $<\delta>$ value at 14 hPa pressure level was about -35 per meg over Japan (Figure 1 in Ishidoya et al., 2013), while only -11 per meg in the tropical stratosphere at the same pressure level.

[p13, L16-17] As described before, the magnitude of gravitational separation in the equatorial stratosphere is almost one third of that observed in the northern mid-latitude at 14 hPa.

*4) a wider consideration of potential age uncertainties and bias and reassessment of the difference between CO2 and SF6-derived mean aged and whether or not it is indeed significant.*

Error estimations of mean age of air derived from $CO_2$ and $SF_6$ have been revised comprehensively as follows. We have revised values of uncertainties for $CO_2$ and $SF_6$ ages as shown in Table 1 and Figure 6 and 8.

[p8, L6-10] Because the uncertainties associated with the best-fit curves will cause an error in the age estimation, we estimated the confidence intervals for the best-fit curves as the root mean squares of CONTRAIL data deviations from the curves, which were 0.65 μmol mol$^{-1}$ and 0.14 pmol mol$^{-1}$ for $CO_2$ and $SF_6$, respectively. These values were comparable or larger than the uncertainties of the mole fraction analyses. How the propagation of uncertainty impacts the age estimation will be discussed later.

[p9, L16-21] Considering the uncertainties associated with the $CO_2$ and $SF_6$ mole fraction measurements and the tropical tropospheric records, overall uncertainties of $CO_2$ and $SF_6$ ages were estimated by the following procedure. At first, normal pseudo random numbers multiplied by 1σ were added to the observed mole fraction data. The same procedure was applied to the tropical tropospheric records. Then, the age calculation procedure described above was repeated for 1000 different sets of random numbers, and the standard deviations of ages were calculated. The overall uncertainties in the estimated ages are also shown in Table 1.

*5) the modelling part of the manuscript is a nice addition, but currently rather inconclusive as none of the tested scenarios seems to be able to match the observations.*

As the referee pointed out, our 2-D model results did not show a good reproducibility for the mean age and gravitational separation by assuming a specific scenario. Because the model study of gravitational separation in the stratosphere has just begun, further investigation will be needed. In this study, we focused on the different sensitivities between gravitational separation and mean age of air to possible perturbations of the mean meridional circulation and the eddy diffusion. For the better understanding of gravitational separation, we also started 3-D model simulations. We have added some sentences at the last of section 3-4 as follows.

[p15, L19-22] Our two-dimensional model results could not reproduce the observed vertical profiles of mean age of air and gravitational separation by assuming a specific scenario, as shown in Figure 8. In order to extend our study on gravitational separation, a three-dimensional model study is needed. It is expected that a three-dimensional model incorporated with the molecular diffusion process can be constrained by gravitational separation data.

*More specific points can be found below.*
*P1, l15 It is common to open with a statement on the motivation for and wider context of the work presented. I encourage the authors to adopt this practice as it also tends to improve readability.*

We have added some sentences to abstract as follows.

[p1, L15-17] The gravitational separation of major atmospheric trace gases, in addition to the age of air, would provide additional useful information about the stratospheric circulation. However, observations of the age of air and gravitational separation are still geographically sparse, especially in the tropics. In order to address this issue,...

*P1, l17 The concept of the mean age of air needs some introduction here. P1, l20 Should this be 29 km instead of 25?*

*P1, l30 Again, the concept of the age of air is not introduced at all.*

We have revised the sentence as follows.

[p1, l23] ... and then rapidly from there up to 29 km.

We have added the sentence of introduction for mean age as follows.

[p1, L32-p2, L1] The mean age of air, defined as the mean transit time of air parcels in the stratosphere, provides important information about various stratospheric transport processes.

*P2, l1 There is also the more recent work of Engel et al., ACP, 2017 extending the balloon-based trend as well as Stiller et al., ACP, 2017 who show that the observed latitudinal and vertical patterns of the decadal changes of age of air in the lower to middle stratosphere during 2002–2012 are predominantly caused by a southward shift of the circulation pattern of about 5 degrees*

*P2, l2 Only in the region of constant mean age above 25 km. In fact, both Diallo et al., ACP, 2012 and Boenisch et al., ACP, 2011 found indications for changes in the region below 25 km.*

*P2, l8-10 Ploeger et al., 2015 demonstrated that both mixing and residual circulation impact on ageing throughout the stratosphere.*

We have added many references in this page including more resent studies (Engel et al., 2017; Bönisch et al., 2011; Diallo et al., 2012; Linz et al., 2017; Garny et al., 2014; Ploeger et al., 2015). We have also added some sentences for the recent knowledge about changes of mean age as follows.

[p2, L9-14] Bönisch et al. (2011) suggested that the increased upwelling in the tropics after 2000 enhanced the lower stratospheric transport from the tropics into the extra-tropics. From an analysis of the ERA-Interim dataset, Diallo et al. (2012) also showed a negative trend over the 1989–2010 period in the lower stratosphere below 25 km. Linz et al. (2017) discussed the strength of the meridional overturning circulation of the stratosphere by using satellite observation data of $SF_6$ and $N_2O$, and suggested that a mesospheric $SF_6$ loss is important for age estimation using $SF_6$ mole fraction in the upper layer.

*P2, l20-22 This is not correct. There is substantially more literature on air sampling in those regions: Volk et al., 1996, Schauffler et al., 1998, Tuck et al., 2004, Kaiser et al., 2006, Laube et al., 2010, and Brinckmann et al., 2012 to name but a few. It would be good to see how the results, in particular the mean ages, compare.*

We have corrected sentences and added many references as follows.

[p2, L27-31] Air sampling has been carried out in the low latitude stratosphere (Volk et al., 1996; Patra et al., 1997; Schauffler et al., 1998; Andrews et al., 2001; Kaiser et al., 2006; Laube et al., 2010; Brinckmann et al.,

2012). However, balloon measurements in the low latitudes, especially to high altitudes (~30 km), have not been conducted as often as those in the middle and high latitudes, mainly due to the limited availability of balloon launching facilities.

*P3, l1-5 Were these samples collected during ascent or descent? In either case, can the authors provide evidence for all species measured that no contamination originated from the balloon (in particular if collection occurred during ascent – Helium leaking into samplers is a known problem) or from the payload?*

We have added some sentences to explain our balloon measurement in more detail as follows.
[p3, L16-20] In the present study, air sampling was performed during balloon ascent. In the past, we conducted a number of similar air sampling using a cryogenic sampler with liquid helium (Honda et al., 1996) in which we collected samples during the balloon ascent and descent over Japan; they showed that the outgassing from the balloon and payload had negligibly small impact on the air sample quality even if air sampling was made during the balloon ascent (Morimoto et al., 2009; Nakazawa et al. 2002).

*P3, l20 What was the reason for the improved precision?*

We have added some sentences to explain a reason why our mass spectrometry was improved as follows.
[p4, L4-8] In this study, only the method of sample air flow into the mass spectrometer was modified from the previous method described in Ishidoya et al. (2013). With this modification, only a miniscule amount of sample air split off from an inlet system was transferred to the mass spectrometer through a fused silica capillary. While sample amount used for this method was larger than before, the precision was improved by one order of magnitude.

*P4, l6-7 What range of mole fractions was covered by those primary standards? ECDs are known to be strongly nonlinear detectors, so what methods were employed to en- sure that a) detector responses were corrected and b) there is agreement between calibration scales over the entire range of observed mole fractions?*

We have revised and added some sentences as follows.
[p4, L20-24] We prepared the primary standard gases (3, 5, 10, 30 pmol mol$^{-1}$, respectively) twice in 2001 (2001 scale) and 2002 (2002 scale), and found that the 2001 scale provides higher values by 0.10-0.15 pmol mol$^{-1}$ than the 2002 scale in a range of observed atmospheric $SF_6$ mole fractions. The relationships between the ECD signal and the mole fractions of the primary standard gases were approximated by quadratic equations.
[p4, L30-31] As a result, our calibration scale agreed, to within our measurement precision, with WMO X2006 and NIES scales over the range of mole fractions observed in this study.

*P4, l25 If CO2 mole fractions do indeed change over time, the exact change will be quite influential on the determined age of air. 0.7 μmol mol−1 is equivalent to several months of mean age and it would be good to know whether the authors have tested any potential variability of such a correction from repeats of storage experiments as well as different water vapour contents, both of which are known to be influential factors. A similar question arises for methane – has this been storage-tested at all?*

As the referee pointed out, deterioration effects have large influence on the age determinations. We have corrected uncertainties of our measurements and reassessed error propagations to the age estimations. Some sentences have been added as follows.

[p5, L9-18] Therefore, we carried out a sample storage test for each bottle to correct for the deterioration effect on $CO_2$ mole fraction. The correction amount ranged from 0.0 μmol mol$^{-1}$ to 0.7 μmol mol$^{-1}$, depending on the bottle. This deterioration effects have large influence on the age determination, because the correction of 0.7 μmol mol$^{-1}$ for $CO_2$ mole fraction is equivalent to 0.3 years of age correction. Therefore, the maximum age error was estimated by taking into account this deterioration for each sampler. We did not make a similar storage test for $SF_6$ prior to use; however, the $SF_6$ mole fraction of sample air was reanalyzed one month after the first set of analyses to check for possible changes in the mole fraction during storage period. Since changes in the $SF_6$ mole fraction were found to be 0.01-0.07 pmol mol$^{-1}$, the deterioration of $SF_6$ during the storage period was neglected. Change in the $CH_4$ mole fraction was also found to be within our measurement precision (Morimoto et al., 2009), and the impact of error propagation to the age determination was negligible.

*P5, l4-5 This is interesting and unusual as the upper part of the profile implies a vertical transport barrier in the tropical stratosphere. Has this been observed in other tropical data? Could this be a regional phenomenon or related to the onset of the recent QBO anomaly? Or is there any evidence for impact from convection? Moreover, can the authors explain why CO2 and SF6 should exhibit a similar behaviour in the lower part of the profile when one has a strong seasonal cycle and the other has none? CO2 would surely be expected to show some variant of a tape-recorder signal, unless the authors have evidence to the contrary?*

The detail of reason for the stepwise change in the vertical profiles is not clear at present. The tape recorder signal of the stratospheric water vapor, ozone, and trajectory analyses would be useful for more quantitative study, which has been discussed in another paper (Hasebe et al., submitted to BAMS). We have modified and added some sentences.

[p5, L29-p6, L6] While the physical details of such complicated vertical $CO_2$ and $SF_6$ profiles are unclear, they can be reasonably reproduced by height-dependent upwelling and/or vertical and horizontal mixing. The vertical propagation of the $CO_2$ seasonal cycle from the troposphere will likely influence the vertical distribution of the $CO_2$ mole fraction especially in the tropical lower stratosphere, since the seasonal amplitude in the tropical upper troposphere was observed to be larger than 3.3 μmol mol$^{-1}$ (as will be described later). In this connection, it is worth noting that the stratospheric water vapor observed during the Biak campaign period showed a clear tape recorder signal of similar behavior (Hasebe et al., submitted to BAMS). For a more quantitative study of the transport processes in the TTL and tropical stratosphere, we need to take a multiple prong approach of integrating $CO_2$ and $SF_6$ data with other variables such as water vapor and $O_3$, using assimilated meteorological data and trajectory analyses.

*P5, l13-16 It is not clear what is being compared here. Similar altitudes cannot be compared directly between tropical and mid latitudes as tropopause heights differ substantially. The authors could use alternative coordinates such as height above tropopause, potential temperature or potential vorticity to address this problem. This would also help answering the question of where the TTL actually was during sampling. In addition, any comparison requires the inclusion of the mid-latitude data set, which could for instance be added to Figure 2.*

We have revised text by using potential temperature as follows. As the referee suggested, we added the results observed in the northern mid-latitude in Figure 2.

[p6, L11-17] We also measured vertical distributions of $CO_2$ and $SF_6$ from 15 to 35 km over Hokkaido, Japan (42° 30' N 143° 26' E) in August 2015 (Figure 2), using our traditional cryogenic sampler with liquid He (Nakazawa et al., 1995; Aoki et al., 2003). To compare the mole fractions of mid-stratospheric $CO_2$ and $SF_6$ in the tropics (Biak) with those observed in the northern mid-latitudes (Hokkaido), average mole fractions were calculated at higher altitudes (potential temperatures larger than 600 K). The latitudinal differences in the $CO_2$ and $SF_6$ mole fractions were found to be 5.6 ± 0.9 µmol mol$^{-1}$ and 1.0 ± 0.2 pmol mol$^{-1}$, respectively.

*P6, l9-12 This is not a valid comparison. In the mid-latitudes, 32 km could mean well over 20 km into the stratosphere, whereas in the tropics the tropopause could be around 19 km. Gravitational separation should be compared at least on a per km in the stratosphere basis and the effects of comparing different pressures should also be considered.*

We have corrected a description of comparison of gravitational separation between the tropics and mid-latitude as follows.

[p7, L13-17] Average vertical gradient of <δ> above the tropopause was -3.3 ± 1.2 per meg km$^{-1}$ in the mid-latitude stratosphere (Ishidoya et al., 2013). On the other hand, our result shows that the average vertical gradient of <δ> was only -1.4 ± 0.4 per meg km$^{-1}$ in the tropical stratosphere. The average <δ> value at 14 hPa pressure level was about -35 per meg over Japan (Figure 1 in Ishidoya et al., 2013), while only -11 per meg in the tropical stratosphere at the same pressure level.

*P6, l27-28 Why are only the last 5 years shown of 10 or more are available? Are the fits less good for the earlier years? What is the uncertainty range of those fits and how high are the resulting uncertainties in mean ages? And why does the data in Figures 4 and 5 have no error bars?*

We have evaluated the statistical uncertainties of our curve fitting and showed them in Figures 4 and 5. Some sentences for the uncertainties have been added as follows. Error bars for mole fractions observed in this study were also shown in Figures 4 and 5.

[p8, L6-10] Because the uncertainties associated with the best-fit curves will cause an error in the age estimation, we estimated the confidence intervals for the best-fit curves as the root mean squares of CONTRAIL data deviations from the curves, which were 0.65 µmol mol$^{-1}$ and 0.14 pmol mol$^{-1}$ for $CO_2$ and $SF_6$, respectively. These values were comparable or larger than the uncertainties of the mole fraction analyses. How the propagation of uncertainty impacts the age estimation will be discussed later.

Uncertainty of mean age derived from data fitting was estimated at different paragraph as follows.

[p9, L16-21] Considering the uncertainties associated with the $CO_2$ and $SF_6$ mole fraction measurements and the tropical tropospheric records, overall uncertainties of $CO_2$ and $SF_6$ ages were estimated by the following procedure. At first, normal pseudo random numbers multiplied by 1σ were added to the observed mole fraction data. The same procedure was applied to the tropical tropospheric records. Then, the age calculation procedure described above was repeated for 1000 different sets of random numbers, and the standard deviations of ages were calculated. The overall uncertainties in the estimated ages are also shown in Table 1.

*P7, l11-12 According to the major ground-based networks (AGAGE and NOAA) global SF6 mole fractions did not increase linearly in the last decade. Emissions have been increasing (WMO, 2014) resulting in some curvature of that trend. Have the authors considered the potential impact on mean ages, or, if they cannot resolve these curvatures, the additional uncertainty in their mean ages?*

We have revised ambiguous expressions which may cause misreading our $SF_6$ age estimation as follows. We didn't assume a linear function for the tropospheric $SF_6$ trend, nor did we calculate $SF_6$ age as lag time in this study.

[p8, L21-24] As seen in Figure 5, the $SF_6$ mole fraction shows no clear seasonal cycle in the tropical upper troposphere and its secular increase for the last 10 years can be approximated by a linear function. Therefore, the $SF_6$ age was sometimes reported as the lag time. However, the non-linear increase in the $SF_6$ mole fraction should be considered for a more precise mean age estimation.

*P8, l8 How were these uncertainties calculated?*

The overall uncertainties of our mean ages have been reevaluated, as described before.

*P8, l9-11 Do the authors actually have any data (e.g. temperature or pressure) to confirm where the tropopause exactly was? And what is meant by "secondary tropopause"?*

During the period of our Biak campaign, many types of balloon observations were carried out, including rawinsondes. We have deleted ambiguous expressions about tropopause and revised as follows.

[p9, L28-29] These results suggest that the $CO_2$ and $SF_6$ ages increased by 0.5 – 0.6 yrs from the tropical upper troposphere (approximately 11 ~ 13 km) to the top of the TTL.

*P8, l24-25 Can the authors present any evidence for this claim?*

Since our observation was highly localized in time and does not have a good representativeness, we have weakened our claim as follows.

[p10, L7-9] This would be partly due to the different time and observation location, although the balloon data observed in the equatorial mid-stratosphere are still relatively sparse and not representative in time and space.

*P8, l29 So other observations do not show a step change in age around 24 km. This reemphasizes my earlier question on an explanation for that phenomenon.*

As described above, we have added some sentences to the description of $CO_2$ and $SF_6$ vertical profiles.

*P9, 16 Again, not necessarily relative to the tropopause.*

We have revised the related sentences by using potential temperature. Average values in the middle stratosphere were re-calculated above 600 K PT, as follows.

[p10, L30-p11, L3] As mentioned before, this study shows that the average $CO_2$ and $SF_6$ ages above the height where the potential temperature was greater than 600 K over Biak are 2.4 ± 0.4 and 3.4 ± 0.5 yrs, respectively.

On the other hand, our balloon observation over Japan in 2015 indicated an average $SF_6$ age of $6.8 \pm 0.8$ yrs for the middle stratosphere (potential temperatures larger than 600 K), which is $1.9 \pm 0.9$ yrs larger than the $CO_2$ age of $4.9 \pm 0.3$ yrs (our unpublished data). This result suggests that the difference in the middle stratospheric $CO_2$ and $SF_6$ ages increases with latitude from the tropics (1.0 yrs) to the mid-latitudes (1.9 yrs).

*P9, l17-24 There are several more important and also more recent studies, e.g. Waugh and Hall, 2002 and in particular Ray et al., 2017.*

We have added Hall and Waugh (1998) as a reference for the discussions of age different between $CO_2$ and $SF_6$. Ray et al. (2017) was also added in the discussions of $SF_6$ loss effect as follows.
[p12, L8-9] More recently, Ray et al. (2017) also reported that the $SF_6$ age in the stratosphere must account for a potential influence from the polar vortex air.

*P10, l19-20 I suggest showing the actual correlation in Figure 6. I would certainly say the two quantities do not correlate well around the upper part of the profile.*

We have deleted a sentence about a correlation between ages and gravitational separation and added the following sentence.
[p12, L16-19] As seen in Figure 6, the increases in the $SF_6$ age with increasing height in the upper layer are accompanied by the gravitational separation enhancement. Similar phenomena were also observed in the high latitudes from our previous balloon experiments (Ishidoya et al., submitted to ASL).

*P10, l24-25 This is quite a claim. Are the authors suggesting that the tropical pipe does not pose a significant transport barrier between tropics and extra-tropics?*

We have deleted this sentence and added the sentence about a possibility that gravitational separation is useful for $SF_6$ loss problem as follows.
 [p12, L20-22] Since gravitational separation will be highly enhanced in the upper stratosphere and the mesosphere, there is a possibility that the impact of $SF_6$ loss on the $SF_6$ age in the upper air or in the polar vortex can be evaluated by using gravitational separation data.

*P11, l6-10 It certainly suggests that the molecular mass difference is dominant, but the molecular diffusion flux could still dampen the signal considerably. A mere proportionality does not rule that out.*

We discussed proportionalities between molecular separations and mass number differences, which is a certain kind of signal. The proportionality in gravitational separation and disproportionality in thermal diffusion are surprisingly rigid, and often observed in the firn air. Therefore, this signal has been used to distinguish each other (i.e., gravitational separation or thermal diffusion). We also tested disproportionality in thermal diffusion in the previous study (Ishidoya et al., 2013), and we concluded that we certainly found the gravitational separation in the stratosphere. We have added a sentence to explain this as follows.
[p13, L7-9] Indeed, in a previous study (Ishidoya et al., 2013), we confirmed that the molecular separations of atmospheric major compositions due to the thermal diffusion are not proportional to the mass number differences.

*P11, l16-17 As pointed out earlier the differences in pressures and temperatures at similar altitudes are quite important here and should be discussed.*

We have added a sentence to describe the latitudinal difference at the same pressure level as follows.
[p13, L16-17] As described before, the magnitude of gravitational separation in the equatorial stratosphere is almost one third of that observed in the northern mid-latitude at 14 hPa.

*P12, l8-9 Did the model-simulated CO2 mole fraction include a realistic seasonal cycle?*

To calculate mean age of air and gravitational separation simultaneously, we used $^{44}CO_2$ and $^{45}CO_2$ as ideal clock tracers without seasonal variations. We have revised this sentence as follows.
[p14, L6-8] a 30-year simulation was performed in which $^{44}CO_2$ and $^{45}CO_2$ were monotonically increased at the model surface, without seasonal variations and keeping their mole fraction ratio constant.

*P12, l23-25 Figure 7 does not include a comparison with data from Ishidoya et al., 2013. I suggest either including it in the figure or as quantitative statements in the text.*

We have revised and added some sentences as follows.
[p14, L22-26] Ishidoya et al. (2013) reported that the average $CO_2$ age and $<\delta>$ in the 30-35 km height layer over Japan for JJA were 4.8 ± 0.4 years and -50 ± 19 per meg, respectively. Note that the $CO_2$ age in Ishidoya et al. (2013) was converted to the CONTRAIL-based value in this study. As shown in Figure 7 (c) and (d), the calculated values of $<\delta>$ and the age of air at mid-stratosphere over northern mid-latitudes in JJA are nearly consistent with the results observed over Japan.

*P21&26, figures 2 and 6 Surely the air samples were collected over a range of altitudes? Please add the y-axis uncertainties for all relevant figures.*

We have added vertical error bars in Figure 2 and 6 to show the vertical ranges of air sampling.

*P23&24, figures 4&5 Some of the scatter observed here is very likely due to changing input from the northern and southern hemisphere as the ITCZ moves through that latitudinal band. Can the authors assess the potential age uncertainty resulting for the exact time of their balloon campaign? This effect might even introduce a bias.*

As described above, we evaluated the uncertainties of curve fitting to the CONTRAIL data by calculating their standard deviations of data, which are shown in Figures 4 and 5. Considering the uncertainties of the tropical tropospheric records, including scattering due to possible ITCZ effect, overall uncertainties of $CO_2$ and $SF_6$ ages were estimated by using normal pseudo random numbers multiplied by $1\sigma$ and by adding random error to the tropical tropospheric records. As a result, overall uncertainties become significantly larger than before as shown in Table 1.

---

## Author Comment (AC2) · 25 Oct 2017

Reply to the referee #2

Our point-by-point response is typed in *italics: referee's comments*, roman: authors' response. Text in the revised manuscript is shown in red with page and line numbers of the new revised manuscript in squared brackets.

*This paper extends the investigation in earlier papers, by several of the same authors, of the gravitational separation of various species in the stratosphere, and its possible implications for atmospheric transport. What is novel, and welcome, about this paper is its focus on tropical data. In principle, the paper appears to be worthy of publication, subject to attention to a few issues, outlined in what follows.*
*Gravitational separation does indeed, as the authors claim, provide a new perspective on stratospheric transport, but it is not made very clear just what that perspective is. Some "ad hoc" experiments are illustrated in the 2D model, in which transport parameters are changed, but things would be made much more clear if there were a theoretical exposition of the problem. For example, one could use simplified models to show how separation would manifest itself in the presence of upwelling alone, or vertical eddy diffusion alone. These would not reproduce the real world, but would provide some theoretical baseline to strengthen understanding of what a more complete model shows. One further shortcoming of the model perturbation experiments discussed in section 3.4 is that the tropics are considered in isolation from the rest of the globe. The authors may get better fits with the tropical data by changing parameters (upwelling, mixing) but that could be at the expense of agreement in the extratropics.*
*The relationship with the vast literature on stratospheric transport could be better illustrated by citing some of this literature more extensively than has been done. Some suggestions are outlined in the following.*

We would like to thank the referee for many critical and constructive comments. As the referee suggested, the physical processes of gravitational separation will be clarified by using a simple model, such as one-dimensional model, rather than 2D model. Indeed, we used 1D steady-state model in the previous study (Ishidoya et al., 2013) and we have also developed 1D dynamical diffusion model with vertical eddy diffusion and advection flux. Those were useful to know basic properties of gravitational separation, such as time constant. We would like to publish them elsewhere. In this study, we focused on the different sensitivities between gravitational separation and mean age of air to possible perturbations of the tropical upwelling and the eddy diffusion. Because the model study of gravitational separation in the stratosphere has just begun, further investigation will be needed. For the better understanding of gravitational separation, we also started 3-D model simulations.
We have obtained a lot of knowledge of gravitational separation and the molecular diffusion processes from studies on the polar firn air, because theoretical basis is the same. To make clear the theoretical background of gravitational separation, we have added an equation that is commonly used in firn air study.

We have tried to address all of referee's comments as detailed below.

*Other issues, as they arise in the text (page, line):*
*(1,32): Determination of the BD circulation from age observations has been explicitly discussed in a recent paper by Linz et al. (Nature Geosci., 2017).*

We have added many references to introduction and discussions for mean age, including Linz et al. (2017). We have added some sentences as follows.

[p2, L9-14] Bönisch et al. (2011) suggested that the increased upwelling in the tropics after 2000 enhanced the lower stratospheric transport from the tropics into the extra-tropics. From an analysis of the ERA-Interim dataset, Diallo et al. (2012) also showed a negative trend over the 1989–2010 period in the lower stratosphere below 25 km. Linz et al. (2017) discussed the strength of the meridional overturning circulation of the stratosphere by using satellite observation data of $SF_6$ and $N_2O$, and suggested that a mesospheric $SF_6$ loss is important for age estimation using $SF_6$ mole fraction in the upper layer.

*(2,10): The separate effects of circulation and mixing on age distributions are discussed in Garny et al. (J. Geophys. Res., 2014) and Linz et al. (J. Atmos.Sci., 2016). The strength of the circulation determines the horizontal gradient of age, rather than age itself; both mixing and circulation determine the vertical structure.*

We have added these studies as references as follows.

[p2, L17-19] Therefore, to discuss a change in the mean age estimated using the clock tracer, it is important to separately evaluate the respective effects of mean circulation and mixing processes on the air age (Garny et al., 2014; Ploeger et al., 2015; Linz et al., 2016).

*(5,31): Is it obvious that gravitational separation depends on mass number difference, rather than, say, mass ratio? If this is a theoretical prediction, please describe it.*

We have revised and added some sentences and equation to make clear the theoretical background about the relationship between gravitational separation and mass number difference as follows.

[p6, L28-32] From a theoretical investigation of the molecular diffusion in polar firn air, the magnitude of the gravitational separation is proportional to mass number difference (Etheridge et al., 1996), which can be expressed as,

$$\Delta\delta = \Delta m \times \Delta\delta_0 \tag{2}$$

Here, $\Delta m$ and $\Delta\delta_0$ are the mass number difference and the difference of δ values for $\Delta m=1$, respectively.

*(6,3): The claim that Figure 3 suggests a linear relationship seems a little exaggerated. There are basically only two points (to be sure, there are 3, but two of them are very close together). Empirically, one could fit any number of curves to the data shown. If there is an a priori expectation of a straight line (from theory, or from more extensive observations) then you should say so. Further, the line does not appear to pass through the origin, which surely it should?*

As the referee pointed out, the proportionality between the molecular separation and mass number difference is theoretically expected and usually observed in polar firn air. In accordance with the theoretical equation (2) described above, the regression line in Figure 3 has been changed to pass through the origin. We have also added some sentences about small deviations from the regression line as follows.

[p7, L6-8] It is not clear what caused the small deviations of Δδ from the proportional relationship shown in Figure 3. The thermal diffusion is one of the plausible causes, but its effect on our observational data taken by using our traditional cryogenic sampler was negligibly small (Ishidoya et al., 2013).

*(8, 25) and elsewhere: Nowhere is it acknowledged that the authors are trying to draw conclusions from data that are highly localized in time, and whose representativeness is therefore open to question. So, e,g., the differences with the Brazil data may due to temporal, rather than spatial, variations. In general, the limitations of the temporal sampling should be prominently acknowledged.*

As the referee suggested, we have revised a sentence to acknowledge the limitations as follows.

[p10, L7-9] This would be partly due to the different time and observation location, although the balloon data observed in the equatorial mid-stratosphere are still relatively sparse and not representative in time and space.

*(9,18): The differences between CO2- and SF6-based age calculations are also discussed in, e.g., Hall and Waugh (J. Geophys. Res., 1998), Ray et al. (J. Geophys. Res., 2017), Linz et al. (Nature Geosci., 2017).*

We have revised and added some sentences by adding these references as follows.

[p11, L4-5] The difference in the $CO_2$ and $SF_6$ ages has also been discussed in previous studies (Harnisch et al., 1998; Hall and Waugh, 1998; Strunk et al., 2000, Andrews et al., 2001).

[p12, L8-9] More recently, Ray et al. (2017) also reported that the $SF_6$ age in the stratosphere must account for a potential influence from the polar vortex air.

[p12, L11-14] Linz et al. (2017) compared the MIPAS $SF_6$ age with the $N_2O$ age calculated with the $N_2O$ data from the Global OZone Chemistry And Related trace gas Data records for the Stratosphere (GOZCARDS), and showed that the MIPAS $SF_6$ age is larger than the $N_2O$ age in the tropics. The $CO_2$ and $SF_6$ ages observed in this study are consistent with the $N_2O$ age rather than the MIPAS $SF_6$ age, although the observation period is different.

*(10,1): Since the CO2 seasonal cycle, like the water vapor "tape recorder" signal, propagates into the tropical lower stratosphere as a decaying sinusoid in the vertical, I do not understand why "the age difference should be larger in the lower stratosphere . . .". It depends on time of year of the data being compared.*

As the referee pointed out, our description was incorrect. We have corrected and added some sentences to make this clear as follows.

[p11, L16-26] For an ideal clock tracer that has increased or decreased monotonically in troposphere, $x(\Gamma, t)$ will be a single-valued function of $\Gamma$, which allows us to determine the mean age of air from the clock tracer mole fraction. On the other hand, if the $CO_2$ seasonal cycle is still significantly large at the observation altitude, it is not necessarily guaranteed that $x(\Gamma, t)$ is a single-valued function of $\Gamma$, depending on the season. In such a case, the $CO_2$ age will be underestimated or overestimated, depending on the time of year, and it is difficult to estimate the $CO_2$ age precisely from the mole fraction at that altitude. If that is the case, then the difference between the $CO_2$ and $SF_6$ ages caused by the $CO_2$ seasonal cycle might be significant in the season when the $CO_2$ mole fraction takes seasonal maxima and minima in the upper troposphere and the lower stratosphere. However, our results showed good agreement between the $CO_2$ and $SF_6$ ages in the TTL and the lower stratosphere. This is probably due to the fact that the seasonal $CO_2$ variation in the equatorial upper troposphere takes an intermediate concentration value in February, a level between its maximum and minimum (Sawa et al., 2008).

*(11,15): If it is the mass dependence of Dmi that matters most, could you show us what that dependence is?*

The molecular diffusion coefficient, $D_{mi}$, strongly depends on the atmospheric pressure. Since the mean free path of a specific molecule increases with decreasing pressure, $D_{mi}$ increases rapidly with increasing altitude. We have revised a sentence as follows.

[p13, L12-14] In addition, the separation effect by molecular diffusion is enhanced with increasing altitude due to the rapid increase of the molecular diffusion coefficient, $D_{mi}$.

*(11,25): Given the relatively small variation of T in the stratosphere, does this term matter much in practice?*

As described in this paragraph, we neglected the thermal diffusion flux in our model calculation (i.e. $\alpha_{Ti}$ =0). This term is related with total number density and the atmospheric scale height and derived from the hydrostatic equation (Banks and Kockarts, 1973). Although the effect of temperature variations on the gravitational separation is not evaluated in this study, its effect was included in our calculations and it would be small at least in the stratosphere. Seasonal variations of gravitational separation shown in Figure 7 might include small temperature effect, while the seasonal change of atmospheric circulation would affect dominantly.

*Figure 7: Why are the altitude scales different on the two frames? And can you comment on the negative ages in the northern lower stratosphere?*

We have corrected the altitude scale in Figure 7. Age of air calculated here was adjusted so that the age values at 17 km are equal to $CO_2$ age observed at 17.2 km (0.4 years). Because SOCRATES does not have a good resolution for the tropospheric modeling (1 km for the vertical coordinate), it seems that the vertical differences of the $CO_2$ mole fraction around the tropopause are not resolved. In addition to this, the vertical transport is so fast in troposphere, which resulted that the $CO_2$ mole fraction was almost constant vertically in the tropics. Therefore, the $CO_2$ mole fraction around and just above tropopause in the northern hemisphere seems to be overestimated in our model.

---

## Author Response (AR2)

**Author's reply to the Co-Editor Decision on "Age and gravitational separation of the stratospheric air over Indonesia" by S. Sugawara et al.**

*Reply to the Co-Editor*

Dear Prof. Peter Haynes

Thank you very much for your helpful comments and for giving us the opportunity to revise the manuscript. We have checked the English and corrected many grammatical errors. We hope that our revised manuscript will be now suitable for ACP publication.

Best regards,
Satoshi

**Reply to the referee #1**

We would like to thank the referee for the constructive comments. We have checked the English and corrected many grammatical errors. We hope that our revised manuscript will be now suitable for ACP publication.